# DualAfford: Learning Collaborative Visual Affordance for Dual-gripper Manipulation

**Yan Zhao**[1,6*]   **Ruihai Wu**[1,6*]   **Zhehuan Chen**[1]   **Yourong Zhang**[1]   **Qingnan Fan**[5]
**Kaichun Mo**[3,4]   **Hao Dong**[1,2,6†]
[1]CFCS, CS Dept., PKU   [2]AIIT, PKU   [3]Stanford University   [4]NVIDIA Research
[5]Tencent AI Lab   [6]BAAI
{zhaoyan790,wuruihai,acmlczh,minor_sixth,hao.dong}@pku.edu.cn
fqnchina@gmail.com   kmo@nvidia.com

## Abstract

It is essential yet challenging for future home-assistant robots to understand and manipulate diverse 3D objects in daily human environments. Towards building scalable systems that can perform diverse manipulation tasks over various 3D shapes, recent works have advocated and demonstrated promising results learning visual actionable affordance, which labels every point over the input 3D geometry with an action likelihood of accomplishing the downstream task (*e.g.*, pushing or picking-up). However, these works only studied single-gripper manipulation tasks, yet many real-world tasks require two hands to achieve collaboratively. In this work, we propose a novel learning framework, DualAfford, to learn collaborative affordance for dual-gripper manipulation tasks. The core design of the approach is to reduce the quadratic problem for two grippers into two disentangled yet interconnected subtasks for efficient learning. Using the large-scale PartNet-Mobility and ShapeNet datasets, we set up four benchmark tasks for dual-gripper manipulation. Experiments prove the effectiveness and superiority of our method over baselines.

## 1 Introduction

We, humans, spend little or no effort perceiving and interacting with diverse 3D objects to accomplish everyday tasks in our daily lives. It is, however, an extremely challenging task for developing artificial intelligent robots to achieve similar capabilities due to the exceptionally rich 3D object space and high complexity manipulating with diverse 3D geometry for different downstream tasks. While researchers have recently made many great advances in 3D shape recognition (Chang et al., 2015; Wu et al., 2015), pose estimation (Wang et al., 2019; Xiang et al., 2017), and semantic understandings (Hu et al., 2018; Mo et al., 2019; Savva et al., 2015) from the vision community, as well as grasping (Mahler et al., 2019; Pinto & Gupta, 2016) and manipulating 3D objects (Chen et al., 2021; Xu et al., 2020) on the robotic fronts, there are still huge perception-interaction gaps (Batra et al., 2020; Gadre et al., 2021; Shen et al., 2021; Xiang et al., 2020) to close for enabling future home-assistant autonomous systems in the unstructured and complicated human environments.

One of the core challenges in bridging the gaps is figuring out good visual representations of 3D objects that are *generalizable* across diverse 3D shapes at a large scale and directly *consumable* by downstream planners and controllers for robotic manipulation. Recent works (Mo et al., 2021; Wu et al., 2022) have proposed a novel perception-interaction handshaking representation for 3D objects – *visual actionable affordance*, which essentially predicts an action likelihood for accomplishing the given downstream manipulation task at each point on the 3D input geometry. Such visual actionable affordance, trained across diverse 3D shape geometry (*e.g.*, refrigerators, microwaves) and for a specific downstream manipulation task (*e.g.*, pushing), is proven to *generalize* to novel unseen objects (*e.g.*, tables) and *benefits* downstream robotic executions (*e.g.*, more efficient exploration).

Though showing promising results, past works (Mo et al., 2021; Wu et al., 2022) are limited to single-gripper manipulation tasks. However, future home-assistant robots shall have two hands just like us humans, if not more, and many real-world tasks require two hands to achieve collaboratively. For example (Figure 1), to steadily pick up a heavy bucket, two grippers need to grasp it at two top

---

*Equal contribution
†Corresponding author; Project page: https://hyperplane-lab.github.io/DualAfford

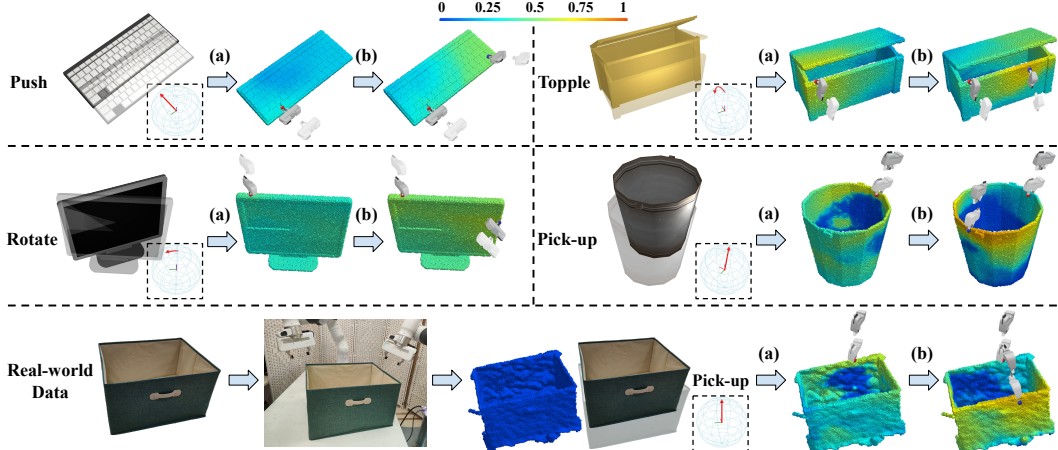

Figure 1: Given different shapes and manipulation tasks (*e.g.*, pushing the keyboard in the direction indicated by the red arrow), our proposed *DualAfford* framework predicts dual collaborative visual actionable affordance and gripper orientations. The prediction for the second gripper (b) is dependent on the first (a). We can directly apply our network to real-world data.

edges and move in the same direction; to rotate a display anticlockwise, one gripper points downward to hold it and the other gripper moves to the other side. Different manipulation patterns naturally emerge when the two grippers collaboratively attempt to accomplish different downstream tasks.

In this paper, we study the dual-gripper manipulation tasks and investigate learning collaborative visual actionable affordance. It is much more challenging to tackle dual-gripper manipulation tasks than single-gripper ones as the degree-of-freedom in action spaces is doubled and two affordance predictions are required due to the addition of the second gripper. Besides, the pair of affordance maps for the two grippers needs to be learned collaboratively. As we can observe from Figure 1, the affordance for the second gripper is dependent on the choice of the first gripper action. How to design the learning framework to learn such collaborative affordance is a non-trivial question.

We propose a novel method *DualAfford* to tackle the problem. At the core of our design, *DualAfford* disentangles the affordance learning problem of two grippers into two separate yet highly coupled subtasks, reducing the complexity of the intrinsically quadratic problem. More concretely, the first part of the network infers actionable locations for the first gripper where there exist second-gripper actions to cooperate, while the second part predicts the affordance for the second gripper conditioned on a given first-gripper action. The two parts of the system are trained as a holistic pipeline using the interaction data collected by manipulating diverse 3D shapes in a physical simulator.

We evaluate the proposed method on four diverse dual-gripper manipulation tasks: pushing, rotating, toppling and picking-up. We set up a benchmark for experiments using shapes from PartNet-Mobility dataset (Mo et al., 2019; Xiang et al., 2020) and ShapeNet dataset (Chang et al., 2015). Quantitative comparisons against baseline methods prove the effectiveness of the proposed framework. Qualitative results further show that our method successfully learns interesting and reasonable dual-gripper collaborative manipulation patterns when solving different tasks. To summarize, in this paper,

- We propose a novel architecture *DualAfford* to learn collaborative visual actionable affordance for dual-gripper manipulation tasks over diverse 3D objects;
- We set up a benchmark built upon SAPIEN physical simulator (Xiang et al., 2020) using the PartNet-Mobility and ShapeNet datasets (Chang et al., 2015; Mo et al., 2019; Xiang et al., 2020) for four dual-gripper manipulation tasks;
- We show qualitative results and quantitative comparisons against three baselines to validate the effectiveness and superiority of the proposed approach.

## 2 RELATED WORK

**Dual-gripper Manipulation.** Many studies, from both computer vision and robotics communities, have been investigating dual-gripper or dual-arm manipulation (Chen et al., 2022; Simeonov et al., 2020; Weng et al., 2022; Chitnis et al., 2020; Xie et al., 2020; Liu & Kitani, 2021; Liu et al., 2022).

Vahrenkamp et al. (2009) presented two strategies for dual-arm planning: J+ and IK-RRT. Cohen et al. (2014) proposed a heuristic search-based approach using a manipulation lattice graph. Ha et al. (2020) presented a closed-loop and decentralized motion planner to avoid a collision. Multi-arm manipulation has also been investigated in various applications: grasping (Pavlichenko et al., 2018), pick-and-place (Shome & Bekris, 2019), and rearrangement (Shome et al., 2021; Hartmann et al., 2021). Our work pays more attention to learning object-centric visual actionable affordance heatmaps for dual-arm manipulation tasks, while previous works focus more on the planning and control sides. Gadre et al. (2021) learns affordances but for interactively part segmentation, they use one gripper to simply hold one articulated part, and use the other gripper to move the other articulated part.

**Visual Affordance Prediction.** Predicting affordance plays an important role in visual understanding and benefits downstream robotic manipulation tasks, which has been widely used in many previous works (Jiang et al., 2021b; Kokic et al., 2017; 2020; Mandikal & Grauman, 2021; Redmon & Angelova, 2015; Wang et al., 2021; Wu et al., 2022). For example, Kokic et al. (2017) used CNN to propose a binary map indicating contact locations for task-specific grasping. Jiang et al. (2021a) proposed the contact maps by exploiting the consistency between hand contact points and object contact regions. Following Where2Act (Mo et al., 2021), we use dense affordance maps to suggest action possibilities at every point on a 3D scan. In our work, we extend by learning two collaborative affordance maps for two grippers that are in deep cooperation for accomplishing downstream tasks.

## 3 PROBLEM FORMULATION

**General Setting.** We place a random 3D object from a random category on the ground, given its partially scanned point cloud observation $O \in \mathbb{R}^{N \times 3}$ and a specific task $l$, the network is required to propose two grippers actions $u_1 = (p_1, R_1)$ and $u_2 = (p_2, R_2)$, in which $p$ is the contact point and $R$ is the manipulation orientation. All inputs and outputs are represented in the camera base coordinate frame, with the z-axis aligned with the up direction and the x-axis points to the forward direction, which is in align with real robot's camera coordinate system.

**Task Formulation.** We formulate four benchmark tasks: pushing, rotating, toppling and picking-up, which are widely used in manipulation benchmarks (Andrychowicz et al., 2017; Kumar et al., 2016; Mousavian et al., 2019; OpenAI et al., 2021) and commonly used as subroutines in object grasping and relocation (Chao et al., 2021; Mahler et al., 2019; Mandikal & Grauman, 2022; Rajeswaran et al., 2018; Zeng et al., 2020). We set different success judgments for difference tasks, and here we describe the pushing task as an example. Task $l \in \mathbb{R}^3$ is a unit vector denoting the object's goal pushing direction. An object is successfully pushed if (1) its movement distance is over 0.05 unit-length, (2) the difference between its actual motion direction $l'$ and goal direction $l$ is within 30 degrees, (3) the object should be moved steadily, *i.e.*, the object can not be rotated or toppled by grippers.

## 4 METHOD

### 4.1 OVERVIEW OF *DualAfford* FRAMEWORK

Figure 2 presents the overview of our proposed *DualAfford* framework. Firstly, we collect large amount of interaction data to supervise the perception networks. Since it is costly to collect human annotations for dual-gripper manipulations, we use an interactive simulator named SAPIEN Xiang et al. (2020). We sample offline interactions by using either a random data sampling method or an optional reinforcement-learning (RL) augmented data sampling method described in Sec. 4.5.

We propose the novel Perception Module to learn collaborative visual actionable affordance and interaction policy for dual-gripper manipulation tasks over diverse objects. To reduce the complexity of the intrinsically quadratic problem of dual-gripper manipulation tasks, we disentangle the task into two separate yet highly coupled subtasks. Specifically, let N denote the point number of the point cloud, and $\theta_R$ denote the gripper orientation space on one point. If the network predicts the two gripper actions simultaneously, the combinatorial search space will be $O\big((\theta_R)^{(N \times N)}\big)$. However, our Perception Module sequentially predicts two affordance maps and gripper actions in a conditional manner, which reduces the search space to $O\big((\theta_R)^{(N+N)}\big)$. Therefore, we design two coupled submodules in the Perception Module: the First Gripper Module $\mathcal{M}_1$ (left) and the Second Gripper Module $\mathcal{M}_2$ (right), and each gripper module consists of three networks (Sec. 4.2).

The training and inference procedures, respectively indicated by the red and blue arrows in Figure 2, share the same architecture but with reverse dataflow directions. For inference, the dataflow direction is intuitive: $\mathcal{M}_1$ proposes $u_1$, and then $\mathcal{M}_2$ proposes $u_2$ conditioned on $u_1$. Although such dataflow guarantees the second gripper plays along with the first during inference, it cannot guarantee the first

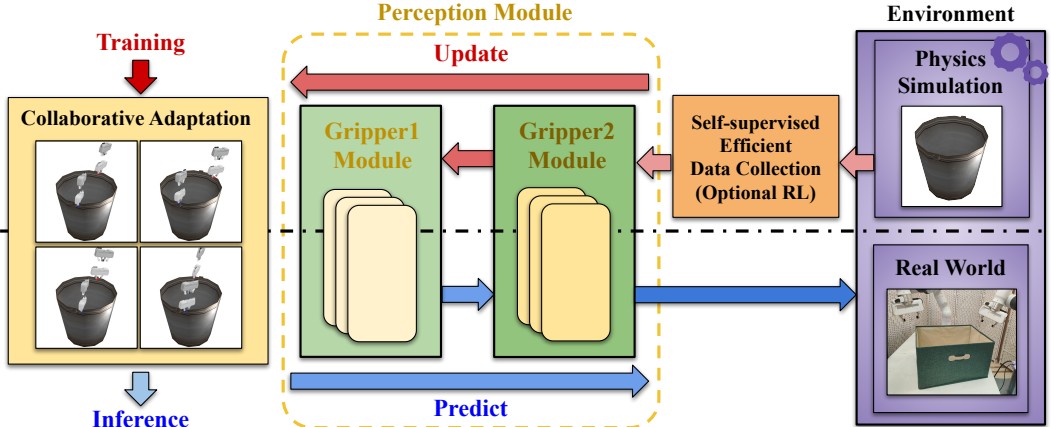

Figure 2: **Our proposed *DualAfford* framework**, first collects interaction data points in physics simulation, then uses them to train the Perception Module, which contains the First Gripper Module and the Second Gripper Module, and further enhances the cooperation between two grippers through the Collaborative Adaption procedure. The training and the inference procedures, as respectively indicated by the red and blue arrows, share the same architecture but with opposite dataflow directions.

gripper's action is suitable for the second to collaborate with. To tackle this problem , for training, we employ the reverse dataflow: $\mathcal{M}_2$ is trained first, and then $\mathcal{M}_1$ is trained with the awareness of the trained $\mathcal{M}_2$. Specifically, given diverse $u_1$ in training dataset, $\mathcal{M}_2$ is first trained to propose $u_2$ collaborative with them. Then, with the trained $\mathcal{M}_2$ able to propose $u_2$ collaborative with different $u_1$, $\mathcal{M}_1$ learns to propose $u_1$ that are easy for $\mathcal{M}_2$ to propose successful collaborations. In this way, both $\mathcal{M}_1$ and $\mathcal{M}_2$ are able to propose actions easy for the other to collaborate with.

Although such design encourages two grippers to cooperate, the two gripper modules are separately trained using only offline collected data, and their proposed actions are never truly executed as a whole, so they are not explicitly taught if their collaboration is successful. To further enhance their cooperation, we introduce the Collaborative Adaptation procedure (Sec. 4.4), in which we execute two grippers' actions simultaneously in simulator, using the outcomes to provide training supervision.

## 4.2 PERCEPTION MODULE AND INFERENCE

To reduce the complexity of the intrinsically quadratic problem and relieve the learning burden of our networks, we disentangle the dual-gripper learning problem into two separate yet coupled subtasks. We design a conditional perception pipeline containing two submodules shown in Figure 3, in which $u_2$ is proposed conditioned on $u_1$ during inference, while $\mathcal{M}_1$ is trained conditioned on the trained $\mathcal{M}_2$ during training. There are three networks in each gripper module: Affordance Network $\mathcal{A}$, Proposal Network $\mathcal{P}$ and Critic Network $\mathcal{C}$. First, as the gripper action can be decomposed into a contact point and a gripper orientation, we design Affordance Network and Proposal Network to respectively predict them. Also, to evaluate whether an action of the gripper is suitable for collaboration, we design Critic Network for this purpose. Below we describe the design of each module.

**Backbone Feature Extractors.** The networks in Perception Module may receive four kinds of input entities or intermediate results: point cloud $O$, task $l$, contact point $p$, and gripper orientation $R$. In different submodules, the backbone feature extractors share the same architectures. We use a segmentation-version PointNet++ (Qi et al., 2017) to extract per-point feature $f_s \in \mathbb{R}^{128}$ from $O$, and employ three MLP networks to respectively encode $l$, $p$, and $R$ into $f_l \in \mathbb{R}^{32}$, $f_p \in \mathbb{R}^{32}$, and $f_R \in \mathbb{R}^{32}$.

### 4.2.1 THE FIRST GRIPPER MODULE

The First Gripper Module contains three sequential networks. Given an object and a task configuration, the Affordance Network $\mathcal{A}_1$ indicates where to interact by predicting affordance map, the Proposal Network $\mathcal{P}_1$ suggests how to interact by predicting manipulation orientations, and the Critic Network $\mathcal{C}_1$ evaluates the per-action success likelihood.

**Affordance Network.** This network $\mathcal{A}_1$ predicts an affordance score $a_1 \in [0, 1]$ for each point $p$, indicating the success likelihood when the first gripper interacts with the point, with the assumption that there exists an expert second gripper collaborating with it. Aggregating the affordance scores, we acquire an affordance map $A_1$ over the partial observation, from which we can filter out low-rated

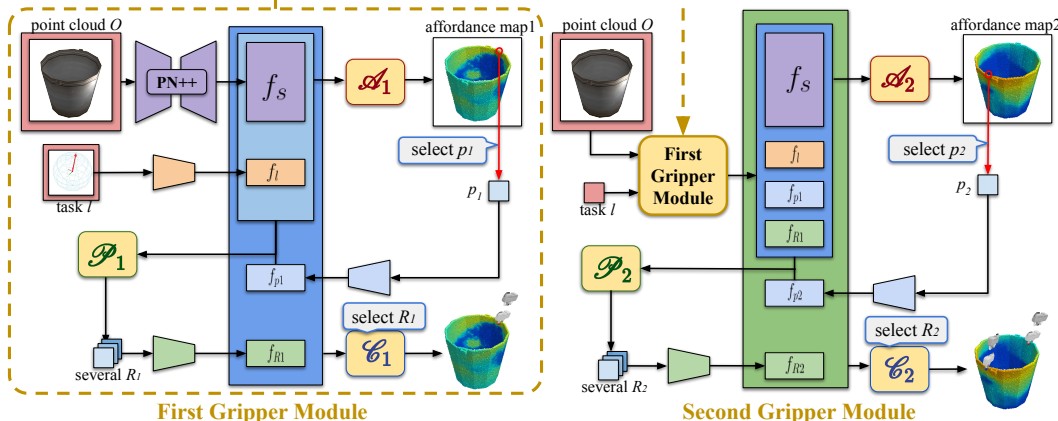

Figure 3: **Architecture details of the Perception Module.** Given a 3D partial scan and a specific task, our network sequentially predicts the first and second grippers' affordance maps and manipulation actions in a conditional manner. Each gripper module is composed of 1) an Affordance Network $\mathscr{A}$ indicating where to interact; 2) a Proposal Network $\mathscr{P}$ suggesting how to interact; 3) a Critic Network $\mathscr{C}$ evaluating the success likelihood of an interaction.

proposals and select a contact point $p_1$ for the first gripper. This network is implemented as a single-layer MLP that receives the feature concatenation of $f_s$, $f_l$ and $f_{p_1}$.

**Proposal Network.** This network $\mathscr{P}_1$ models the distribution of the gripper's orientation $R_1$ on the given point $p_1$. It is implemented as a conditional variational autoencoder (Sohn et al., 2015), where an encoder maps the gripper orientation into a Gaussian noise $z \in \mathbb{R}^{32}$, a decoder reconstructs it from $z$. Implemented as MLPs, they both take the feature concatenation of $f_s$, $f_l$, and $f_{p_1}$ as the condition.

**Critic Network.** This network $\mathscr{C}_1$ rates the success likelihood of each manipulation orientation on each point by predicting a scalar $c_1 \in [0, 1]$. A higher $c_1$ indicates a higher potential for the second gripper to collaboratively achieve the given task. It is implemented as a single-layer MLP that consumes the feature concatenation of $f_s$, $f_l$, $f_{p_1}$ and $f_{R_1}$.

### 4.2.2 THE SECOND GRIPPER MODULE

Conditioned on the first gripper action $u_1 = (p_1, R_1)$ proposed by $\mathscr{M}_1$, $\mathscr{M}_2$ first generates a point-level collaborative affordance $A_2$ for the second gripper and samples a contact point $p_2$. Then, $\mathscr{M}_2$ proposes multiple candidate orientations, among which we can choose a suitable one as $R_2$. The design philosophy and implementations of $\mathscr{M}_2$ are the same as $\mathscr{M}_1$, except that all three networks ($\mathscr{A}_2$, $\mathscr{P}_2$ and $\mathscr{C}_2$) take the first gripper's action $u_1$, i.e., $p_1$ and $R_1$, as the additional input.

### 4.3 TRAINING AND LOSSES

As shown in Figure 2, during inference (indicated by blue arrows), the first gripper predicts actions without seeing how the second gripper will collaborate. To enable the first gripper to propose actions easy for the second to collaborate with, we train the Perception Module in the dataflow direction indicated by red arrows, as described in Sec. 4.1. We adopt the Critic Network $\mathscr{C}_1$ of the first gripper as a bridge to connect two gripper modules. $\mathscr{C}_1$ scores whether an action of the first gripper is easy for $\mathscr{M}_2$ to propose collaborative actions. With the trained $\mathscr{C}_1$, $\mathscr{M}_1$ will propose actions with the assumption that there exists an expert gripper to cooperate with. Therefore, $\mathscr{M}_1$ and $\mathscr{M}_2$ will both learn to collaborate with each other.

**Critic Loss.** It is relatively easy to train the second Critic Network $\mathscr{C}_2$. Given the interaction data with the corresponding ground-truth interaction result $r$, where $r = 1$ means positive and $r = 0$ means negative, we can train $\mathscr{C}_2$ using the standard binary cross-entropy loss. For simplicity, we use $f^{in}$ to denote each network's input feature concatenation, as mentioned in Sec.4.2:

$$\mathscr{L}_{\mathscr{C}_2} = r_j \log\big(\mathscr{C}_2(f^{in}_{p_2})\big) + (1 - r_j)\log\big(1 - \mathscr{C}_2(f^{in}_{p_2})\big). \tag{1}$$

However, for the first Critic Network $\mathscr{C}_1$, since we only know the first gripper's action $u_1 = (p_1, R_1)$, we can not directly obtain the ground-truth interaction outcome of a single action $u_1$. To tackle this problem, given the first gripper's action, we evaluate it by estimating the potential for the second

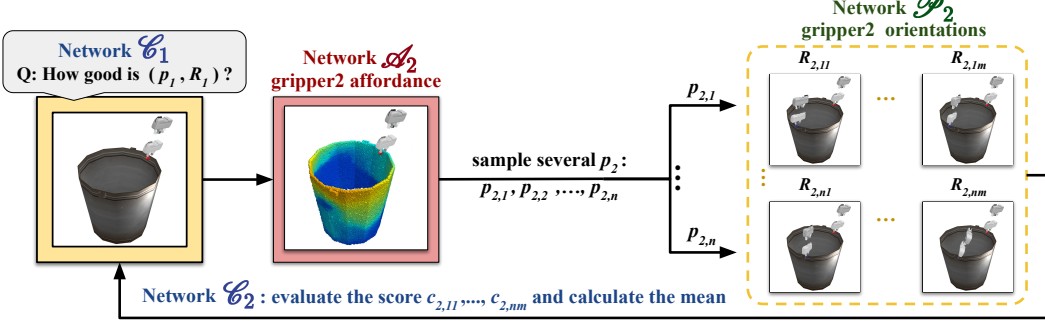

Figure 4: To train $\mathscr{C}_1$ that evaluates how the first action can collaborate with the trained Second Gripper Module $\mathscr{M}_2$, we comprehensively use the trained $\mathscr{A}_2$, $\mathscr{P}_2$ and $\mathscr{C}_2$ of $\mathscr{M}_2$ to provide supervision.

gripper to collaboratively accomplish the given task. As shown in Figure 4, we comprehensively use the trained $\mathscr{A}_2$, $\mathscr{P}_2$ and $\mathscr{C}_2$ of the Second Gripper Module $\mathscr{M}_2$. Specifically, to acquire the ground-truth action score $\hat{c}$ for the first gripper, we first use $\mathscr{A}_2$ to predict the collaborative affordance map $A_2$ and sample $n$ contact points: $p_{2,1}, ..., p_{2,n}$, then we use $\mathscr{P}_2$ to sample $m$ interaction orientations on each contact point $i$: $R_{2,i1}, ..., R_{2,im}$. Finally, we use $\mathscr{C}_2$ to rate the scores of these actions: $c_{2,11}, ..., c_{2,nm}$ and calculate their average value. Thus we acquire the ground-truth score of $\mathscr{C}_1$, and we apply $\mathscr{L}_1$ loss to measure the error between the prediction and the ground-truth:

$$\hat{c}_{p_1} = \frac{1}{nm} \sum_{j=1}^{n} \sum_{k=1}^{m} \mathscr{C}_2\big(f_{p_{2j}}^{in}, \mathscr{P}_2(f_{p_{2j}}^{in}, z_{jk})\big); \qquad \mathscr{L}_{\mathscr{C}_1} = \big|\mathscr{C}_1(f_{p_1}^{in}) - \hat{c}_{p_1}\big|. \tag{2}$$

**Proposal Loss.** $\mathscr{P}_1$ and $\mathscr{P}_2$ are implemented as cVAE (Sohn et al., 2015). For the $i$-th gripper, we apply geodesic distance loss to measure the error between the reconstructed gripper orientation $R_i$ and ground-truth $\hat{R}_i$, and KL Divergence to measure the difference between two distributions:

$$\mathscr{L}_{\mathscr{P}_i} = \mathscr{L}_{geo}(R_i, \hat{R}_i) + D_{KL}\big(q(z|\hat{R}_i, f^{in})||\mathscr{N}(0,1)\big). \tag{3}$$

**Affordance Loss.** Similar to Where2Act (Mo et al., 2021), for each point, we adopt the 'affordance' score as the expected success rate when executing action proposals generated by the Proposal Network $\mathscr{P}$, which can be directly evaluated by the Critic Network $\mathscr{C}$. Specifically, to acquire the ground-truth affordance score $\hat{a}$ for the $i$-th gripper, we sample $n$ gripper orientations on the point $p_i$ using $\mathscr{P}_i$, and calculate their average action scores rated by $\mathscr{C}_i$. We apply $\mathscr{L}_1$ loss to measure the error between the prediction and the ground-truth affordance score on a certain point:

$$\hat{a}_{p_i} = \frac{1}{n} \sum_{j=1}^{n} \mathscr{C}_i\big(f_{p_i}^{in}, \mathscr{P}_i(f_{p_i}^{in}, z_j)\big); \qquad \mathscr{L}_{\mathscr{A}_i} = \big|\mathscr{A}_i(f_{p_i}^{in}) - \hat{a}_{p_i}\big|. \tag{4}$$

### 4.4 COLLABORATIVE ADAPTATION PROCEDURE

Although the above training procedure can enable two gripper modules to propose affordance and actions collaboratively, their collaboration is limited, because they are trained in a separate and sequential way using only offline collected data, without any real and simultaneous executions of proposed actions. To further enhance the collaboration between the two gripper modules, we introduce the Collaborative Adaptation procedure, in which the two modules are trained in a simultaneous manner using online executed and collected data, with loss functions the same as in Sec. 4.3. In this procedure, the proposed dual-gripper actions are simultaneously executed in the simulator, using interaction outcomes to update the two gripper modules. In this way, the two gripper modules can better understand whether their proposed actions are successful or not as they are aware of interaction results, and thus the two separately trained modules are integrated into one collaborative system.

### 4.5 OFFLINE DATA COLLECTION

Instead of acquiring costly human annotations, we use SAPIEN (Xiang et al., 2020) to sample large amount of offline interaction data. For each interaction trial with each object, we sample two gripper actions $u_1, u_2$, and test the interaction result $r$. We define a trial to be positive when: (1) the two grippers successfully achieve the task, *e.g.*, pushing a display over a threshold length without rotating it; (2) the task can be accomplished only by the collaboration of two grippers, *i.e.*, when we replay each gripper action without the other, the task can not be achieved. We represent each interaction data as $(O, l, p_1, p_2, R_1, R_2) \rightarrow r$, and balance the number of positive and negative interactions. Here we introduce two data collection methods: random and RL augmented data sampling.

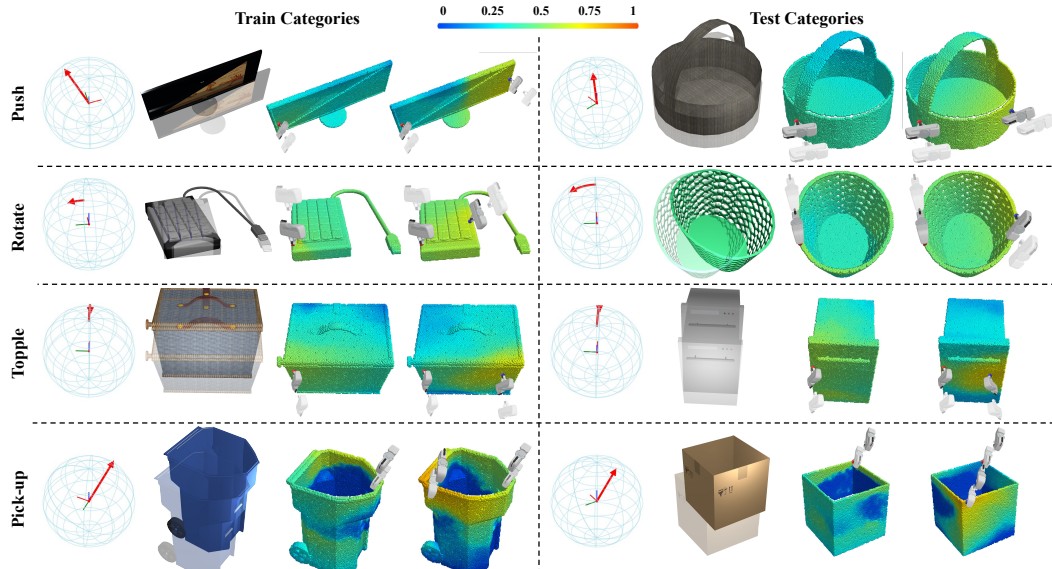

Figure 5: Qualitative results of Affordance Networks. In each block, we respectively show (1) task represented by a red arrow, (2) object which should be moved from transparent to solid, (3) the first affordance map predicted by $\mathscr{A}_1$, (4) the second affordance map predicted by $\mathscr{A}_2$ conditioned on the first action. Left shapes are from training categories, while right shapes are from unseen categories.

**Random Data Sampling.** We can efficiently sample interaction data by parallelizing simulation environments across multiple CPUs. For each data point, we first randomly sample two contact points on the object point cloud, then we randomly sample two interaction orientations from the hemisphere above the tangent plane around the point, and finally test the interaction result.

**RL Augmented Data Sampling.** For tasks with complexity, such as picking-up, it is nearly impossible for a random policy to collect positive data. To tackle this problem, we propose the RL method. We first leverage Where2Act (Mo et al., 2021) to propose a prior affordance map, highlighting where to grasp. After sampling two contact points, we use SAC (Haarnoja et al., 2018) with the manually designed dense reward functions to efficiently predict interaction orientations.

## 5 EXPERIMENTS

### 5.1 RESULTS AND ANALYSIS

We perform large-scale experiments under four dual-gripper manipulation tasks, and set up three baselines for comparisons. Results prove the effectiveness and superiority of our proposed approach.

### 5.2 ENVIRONMENT SETTINGS AND DATASET

We follow the environment settings of Where2Act (Mo et al., 2021) except that we use two Franka Panda Flying grippers. We conduct our experiments on SAPIEN (Xiang et al., 2020) simulator with the large-scale PartNet-Mobility (Mo et al., 2019) and ShapeNet (Chang et al., 2015) dataset. To analyze whether the learned representations can generalize to novel unseen categories, we reserve some categories only for testing. See Supplementary Sec. C for more details.

### 5.3 EVALUATION METRICS, BASELINES AND ABLATION

**Evaluation Metrics.** To quantitatively evaluate the action proposal quality, we run interaction trials in simulation and report sample-success-rate (Mo et al., 2021), which measures the percentage of successful interactions among all interaction trials proposed by the networks.

**Baselines.** We compare our approach with three baselines and one ablated version: (1) A random approach that randomly selects the contact points and gripper orientations. (2) A heuristic approach in which we acquire the ground-truth object poses and hand-engineer a set of rules for different tasks. For example, for the picking-up task, we set the two contact points on the objects' left and right top edges and set the two gripper orientations the same as the given picking-up direction. (3) M-Where2Act: a dual-gripper Where2Act (Mo et al., 2021) approach. While Where2Act initially

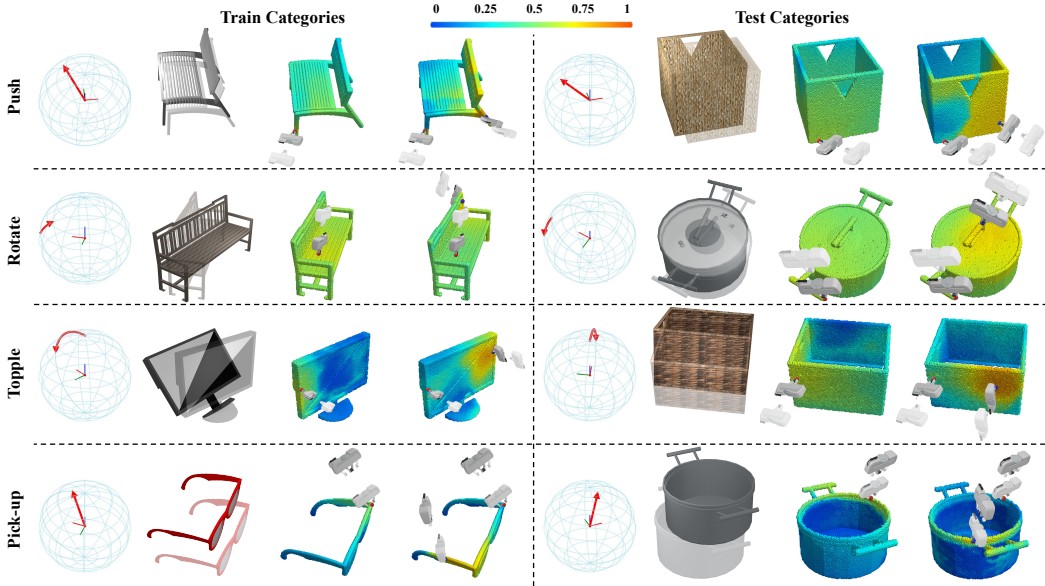

Figure 6: The per-point action scores predicted by Critic Networks $\mathscr{C}_1$ and $\mathscr{C}_2$. In each result block, from left to right, we show the task, the input shape, the per-point success likelihood predicted by $\mathscr{C}_1$ given the first gripper orientation, and the per-point success likelihood predicted by $\mathscr{C}_2$ given the second gripper orientation, conditioned on the first gripper's action.

considers interactions for a single gripper, we adapt it as a baseline by modifying each module in Where2Act to consider the dual grippers as a combination, and assign a task $l$ to it as well. (4) Ours w/o CA: an ablated version of our method that removes the Collaborative Adaptation procedure.

Figure 5 presents the dual affordance maps predicted by our Affordance Networks $\mathscr{A}_1$ and $\mathscr{A}_2$, as well as the proposed grippers interacting with the high-rated points. We can observe that: (1) the affordance maps reasonably highlight where to interact (*e.g.*, for picking-up, the grippers can only grasp the top edge); (2) the affordance maps embody the cooperation between the two grippers (*e.g.*, to collaboratively push a display, the two affordance maps sequentially highlight its left and right half part, so that the display can be pushed steadily.) Besides, we find that our method has the ability to generalize to novel unseen categories.

In Figure 6, we additionally visualize the results of Critic Networks $\mathscr{C}_1$ and $\mathscr{C}_2$. Given different gripper orientations, the Critic Networks propose the per-point action scores over the whole point cloud. We can observe that our network is aware of the shape geometries, gripper orientations and tasks. For example, in the Rotate-Train-Categories block, the first map highlights a part of chair surface since the first gripper is downward, and the second map accordingly highlights the chair back on the other side given the second-gripper orientation, which collaboratively ensures the chair is rotated clockwise. It is noteworthy that in the first map the chair surface has higher scores than the arm, because the chair tends to skid when selecting the arm as a fulcrum for rotation.

Figure 7 (a) visualizes the diverse collaborative actions proposed by Proposal networks $\mathscr{P}_1$ and $\mathscr{P}_2$ on an example display. Our networks can propose different orientations on the same points.

Table 1: Baseline comparison on the sample-success-rate metric.

| | Train Categories | | | | Test Categories | | | |
|---|---|---|---|---|---|---|---|---|
| | pushing | rotating | toppling | picking-up | pushing | rotating | toppling | picking-up |
| Random | 7.40 | 10.40 | 6.40 | 3.00 | 3.20 | 9.00 | 3.00 | 6.00 |
| Heuristic | 32.40 | 24.20 | 54.00 | 31.93 | 25.80 | 21.80 | 38.00 | 37.90 |
| M-Where2Act | 28.00 | 15.67 | 36.60 | 5.00 | 23.40 | 10.67 | 25.60 | 13.80 |
| Ours w/o CA | 35.87 | 17.53 | 56.00 | 28.87 | 34.67 | 15.33 | 39.67 | 38.33 |
| Ours | **48.76** | **33.73** | **65.53** | **40.33** | **42.93** | **35.07** | **41.80** | **54.33** |

Table 1 presents the sample-success-rate of different methods over the four challenging tasks. We can see that our method outperforms three baselines over all comparisons.

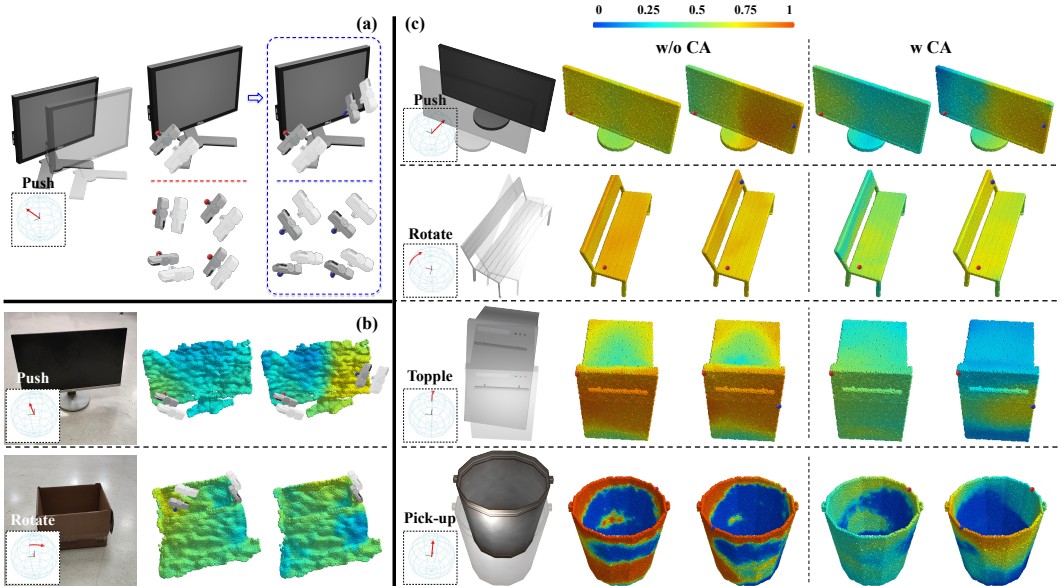

Figure 7: (a) The diverse and collaborative actions proposed by the Proposal Networks $\mathscr{P}_1$ and $\mathscr{P}_2$. (b) The promising results testing on real-world data. (c) The actionable affordance maps of the ablated version that removes the Collaborative Adaptation procedure (left) and ours (right).

For **the heuristic baseline**, it gains relatively high numbers since it proposes actions with the ground-truth object poses and orientations. However, the inter- and intra-category shape geometries are exceptionally diverse, and we can not expect the hand-engineered rules to work for all shapes. Moreover, in the real world, this approach needs more effort to acquire ground-truth information.

For **M-Where2Act**, it learns the dual contact points and orientations as a combination and has worse performance. In comparison, our method disentangles the collaboration learning problem and reduces the complexity. Besides, M-Where2Act consumes nearly quadratic time to give proposals for the reason that it has to query the affordance scores of all the $n * n$ pair combinations of $n$ points.

For **Ours w/o CA**, this ablated version of our method shows that the Collaborative Adaption procedure helps boost the performance. Figure 7 (c) visualizes the affordance maps without (left) and with (right) Collaborative Adaption procedure. We find that the affordance maps become more refined. For example, to push the display, the affordance scores of the base become lower since it is difficult to interact with; to collaboratively topple the dishwasher, in the second affordance map, the left front surface receives lower scores while the right maintains relatively higher.

Table 2 shows the success rate of the Random Data Sampling and RL Augmented Data Sampling method. The RL method significantly improves data collection efficiency on each object category.

Figure 1 and Figure 7 (b) show qualitative results that our networks can directly transfer the learned affordance to real-world data. We show more real-robot experiments in supplementary Sec. A.

Table 2: The success rate of data collection in the picking-up task.

| | Train Categories | | | | | | Test Categories | | | |
|---|---|---|---|---|---|---|---|---|---|---|
| | eyeglasses | bucket | trash can | pliers | basket | display | box | kitchen pot | scissors | laptop |
| Random-Sampling | 0.06 | 0.12 | 0.04 | < 0.01 | 0.09 | 0.03 | 0.03 | 0.06 | < 0.01 | < 0.01 |
| RL-Sampling | **6.12** | **9.65** | **5.26** | **5.79** | **6.41** | **9.78** | **5.12** | **9.18** | **6.38** | **7.13** |

## 6 CONCLUSION

In this paper, we proposed a novel framework *DualAfford* for learning collaborative actionable affordance for dual-gripper manipulation over diverse 3D shapes. We set up large-scale benchmarks for four dual-gripper manipulation tasks using the PartNet-Mobility and ShapeNet datasets. Results proved the effectiveness of the approach and its superiority over the three baselines.

**Acknowledgements.**  Yan Zhao, Ruihai Wu, Zhehuan Chen, Yourong Zhang, Hao Dong were supported by National Natural Science Foundation of China (No. 62136001).

**Ethics Statements.**  Our project helps build future home-assistant robots. The visual affordance predictions can be visualized and analyzed before use, which is less vulnerable to potential dangers. The training data used may introduce data bias, but this is a general concern for similar methods. We do not see any particular major harm or issue our work may raise up.

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

APPENDIX

# A  REAL-ROBOT SETTINGS AND EXPERIMENTS

For real-robot experiments, we set up two Franka Panda robots, and place various target objects like boxes, buckets and displays in front of them. One RealSense i435 camera is mounted behind the two robots. The camera captures partial 3D point cloud data as inputs to our learned models.

We control the two robots using Robot Operating System (ROS) (Quigley et al., 2009). The two robots are programmed to execute the actions proposed by our method. We use MoveIt! (Chitta et al., 2012) for the motion planning.

In Figure 8, we present some promising results by directly testing our pre-trained model over real-world scans. We observe that our model has a good generalization capability.

Please check our video for more results.

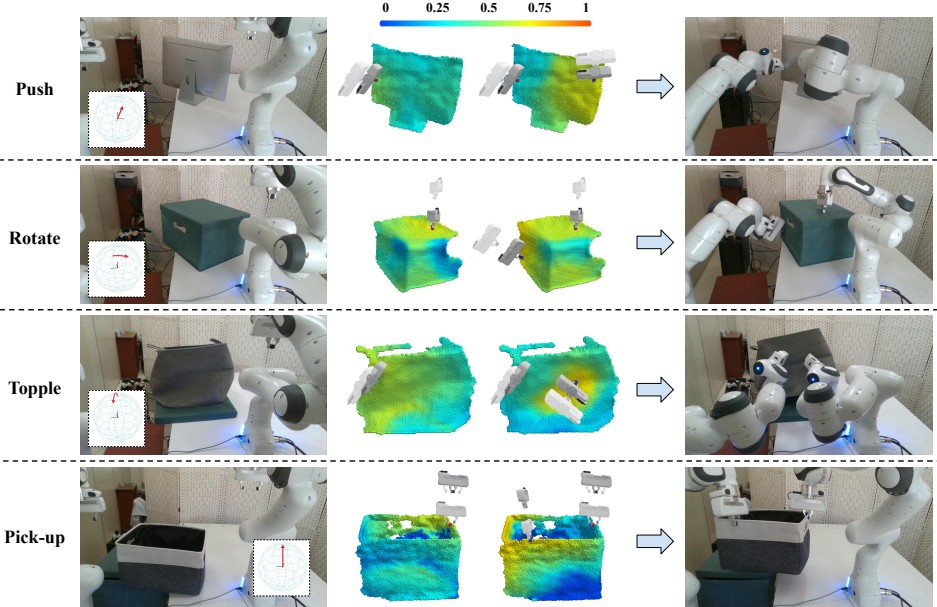

Figure 8: We present some promising results by directly testing our model on real-world scans. In each block, from left to right, we respectively show the task represented by a red arrow, the input shape, the actionability heatmap predicted by network $\mathscr{A}_1$, and the actionability heatmap predicted by network $\mathscr{A}_2$ conditioned on the action of the first gripper. Please check our video for more results.

# B  MORE DETAILS ABOUT EXPERIMENT SETTING AND TASK FORMULATION

## B.1  EXPERIMENT SETTING

All inputs and outputs are represented in the camera base coordinate frame, with the z-axis aligned with the up direction and the x-axis points to the forward direction, which is in align with real robot's camera coordinate system.

## B.2  TASK FORMULATION

We formulate four benchmark tasks: pushing, rotating, toppling and picking-up. We have explained the pushing task in the main paper, and here we describe the remaining tasks:

- For rotating, $l = \theta \in \mathbb{R}$ denotes the object's goal rotation angle on the horizontal plane. $\theta > 0$ denotes the object is rotated anticlockwise, $\theta < 0$ denotes the object is rotated clockwise. An object is successfully rotated if its actual rotation degree $\theta'$ is over 10 degrees, its rotation direction is the same as $\theta$ (*i.e.*, $\theta * \theta' > 0$), and the difference between $\theta'$ and $\theta$ is less than 30 degrees.

- For toppling, $l \in \mathbb{R}^3$ is a unit vector denoting the object's goal toppling direction. An object is successfully toppled if its toppling angle is over 10 degrees, the difference between its actual toppling direction $l'$ and $l$ is less than 30 degrees, and the object can not be rotated.

- For picking up, $l \in \mathbb{R}^3$ is a unit vector denoting the object's goal motion direction. An object is successfully picked up if its movement height is over 0.05 unit-length, the difference between its actual motion direction $l'$ and $l$ is less than 45 degrees, and the object should keep steady.

## C   DATA STATISTICS

We perform our experiments using the large-scale PartNet-Mobility(Mo et al., 2019) dataset and ShapeNet(Chang et al., 2015) dataset. We select object categories suitable for each dual-gripper manipulation task and remove objects that can be easily manipulated by a single gripper (*e.g.,* small objects like USB drives that are easy to to push). To analyze whether the learned representations can generalize to novel unseen categories, we randomly select some categories for training and reserve the rest categories only for testing. We further split the training set into training shapes and testing shapes, and only used the training shapes from the training categories to train our framework.

We summarize the shape counts from the training categories in Table 3, and the shape counts from the novel unseen categories in Table 4. In Table 3, the number before slash is the training shape counts from the training categories, and the number after slash is the testing shape counts from the training categories. We label the categories from the ShapeNet dataset with $\star$, and the remaining categories are from PartNet-Mobility dataset.

Figure 9 visualizes one example for each object category from the dataset we use in our experiment.

Table 3: **The shape counts from the training categories.** We further spilt the training set into training shapes (before slash) and testing shapes(after slash), and only used the training shapes from the training categories to train our framework.

| pushing | All | | Box | Dishwasher | Display | Bench$^\star$ | Keyboard$^\star$ | |
|---|---|---|---|---|---|---|---|---|
| | 272 / 92 | | 21 / 7 | 35 / 12 | 28 / 8 | 141 / 48 | 47 / 17 | |
| rotating | All | | Box | Dishwasher | Display | Bench$^\star$ | Keyboard$^\star$ | |
| | 272 / 92 | | 21 / 7 | 35 / 12 | 28 / 8 | 141 / 48 | 47 / 17 | |
| toppling | All | | Box | Bucket | Dishwasher | Display | Bench$^\star$ | Bowl$^\star$ |
| | 383 / 126 | | 21 / 7 | 23 / 7 | 35 / 12 | 28 / 8 | 141 / 48 | 135 / 44 |
| picking-up | All | | Eyeglasses | Bucket | Trash Can | Pliers | Basket$^\star$ | Display |
| | 215 / 85 | | 49 / 16 | 17 / 8 | 42 / 16 | 19 / 6 | 66 / 31 | 22 / 8 |

## D   MORE DETAILS ABOUT PERCEPTION MODULE

The Perception Module has two submodules: the First Gripper Module and the Second Gripper Module. Below we will explain the network architecture details of the First Gripper Module, which consists of four input encoders and three parallel output networks.

Regard the four input encoders

- The partial point cloud $O \in \mathbb{R}^{N \times 3}$ is fed through a segmentation-version PointNet++ (Qi et al., 2017) to extract the per-point feature $f_s \in \mathbb{R}^{128}$.

Table 4: **The shape counts from the novel unseen categories.**

| | All | Kitchen Pot | Toaster | Basket⋆ | |
|---|---|---|---|---|---|
| pushing | | | | | |
| | 164 | 25 | 25 | 114 | |
| rotating | All | Kitchen Pot | Toaster | Basket⋆ | |
| | 164 | 25 | 25 | 114 | |
| toppling | All | Kitchen Pot | Toaster | Basket⋆ | |
| | 164 | 25 | 25 | 114 | |
| picking-up | All | Box | Kitchen Pot | Scissors | Laptop |
| | 84 | 27 | 19 | 16 | 22 |

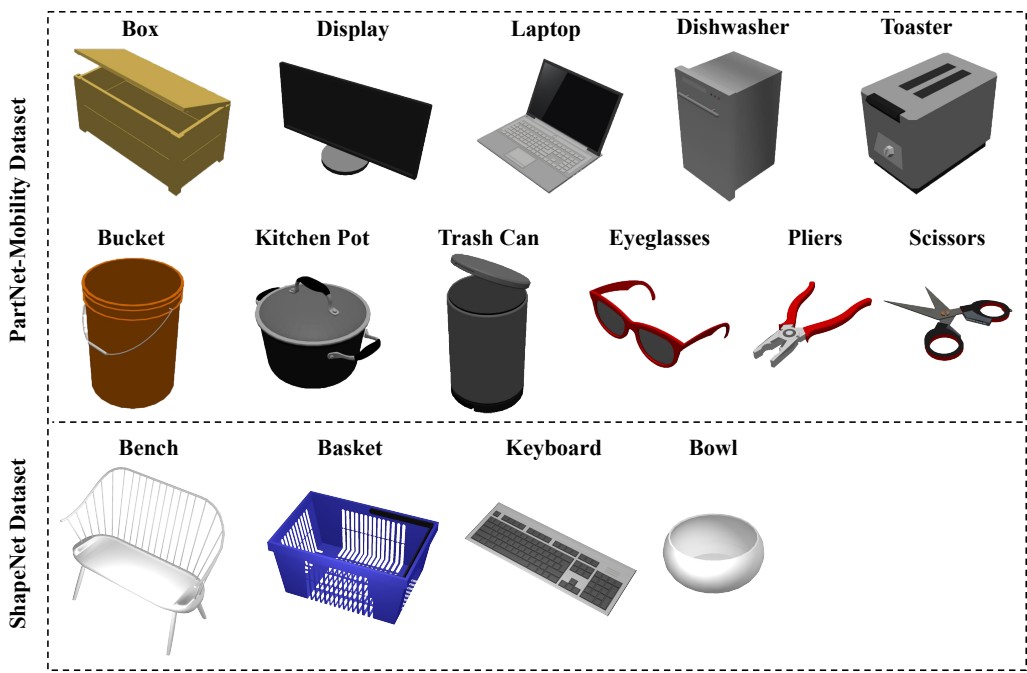

Figure 9: We visualize one example for each object category used in our work.

- The task $l \in \mathbb{R}^3$ (for pushing, toppling, picking-up) or $l \in \mathbb{R}^1$ (for rotating) is fed through a fully connected layer ($3 \to 32$ or $1 \to 32$) to extract the feature $f_p \in \mathbb{R}^{32}$.

- The contact point $p \in \mathbb{R}^3$ is fed through a fully connected layer ($3 \to 32$) to extract the feature $f_p \in \mathbb{R}^{32}$.

- The manipulation orientation $R \in \mathbb{R}^6$ is represented by the first two orthonormal axes in the $3 \times 3$ rotation matrix, following the proposed 6D-rotation representation in (Zhou et al., 2019). We fed the orientation through a fully connected layer ($3 \to 32$) to extract the feature $f_R \in \mathbb{R}^{32}$.

Regarding the three output networks

- The affordance network of the first gripper $\mathscr{A}_1$ concatenates the feature $f_s$, $f_l$, and $f_{p_1}$ to form a high-dimensional vector, which is fed through a 2-layer MLP($192 \to 128 \to 1$) to predict a per-point affordance score $a_1 \in [0,1]$.

- The proposal network of the first gripper $\mathscr{P}_1$ is implemented as a conditional variational autoencoder(cVAE) (Sohn et al., 2015), which takes the feature concatenation of $f_s$, $f_l$, and $f_{p_1}$ as conditions. The encoder is implemented as a 3-layer MLP ($224 \to 128 \to 32 \to 32$) that takes the orientation feature $f_{R_1}$ and the given conditions to estimate a Gaussian distribution ($\mu \in \mathbb{R}^{32}, \sigma \in \mathbb{R}^{32}$). The decoder is implemented as a 2-layer MLP

$(224 \rightarrow 128 \rightarrow 6)$ that takes the concatenation of the sampled Gaussian noise $z \in \mathbb{R}^{32}$ and the given conditions to recover the manipulation orientation $R_1$.

- The critic network of the first gripper $\mathscr{C}_1$ concatenates the feature $f_s$, $f_l$, $f_{p_1}$ and $f_{R_1}$ to form a high-dimensional vector, which is fed through a 2-layer MLP$(224 \rightarrow 128 \rightarrow 1)$ to predict the success likelihood $c_1 \in [0, 1]$.

For the Second Gripper Module, the architecture of its input encoders and output networks are the same as those in the First Gripper Module, except that they take the first gripper's action *i.e.*, $p_1$ and $R_1$, as the additional input. Therefore, the input dimension of the three parallel output networks should be added with 64, while other feature dimensions remain unchanged.

The hyper-parameters used to train the Perception Module are as following: 32 (batch size); 0.001 (learning rate of the Adam optimizer (Kingma & Ba, 2014)) for all three networks.

## E  MORE DETAILS ABOUT RL AUGMENTED DATA SAMPLING

### E.1  THE RL POLICY FOR INTERACTIVE TRAJECTORY EXPLORATION

It is too difficult for random policies to collect positive data for the picking-up tasks, since the objects and tasks are too complex. Therefore, we train an RL policy using SAC (Haarnoja et al., 2018) replacing the random policy to collect enough positive dual-gripper interactions. We access the ground-truth state information of the simulation environment, such as object poses and orientations, and utilize them for the RL training.

**State Space.**  The RL state is the concatenation of the object position $p_o \in \mathbb{R}^3$, the object orientation $q_o \in \mathbb{R}^4$, and the two contact points $p_1, p_2 \in \mathbb{R}^3$. In order to select contact points in more reasonable positions, we apply the pre-trained Where2Act (Mo et al., 2021) pulling affordance on the observation and randomly sample two different points with the top 10% affordance scores. Since the grippers have only one chance to interact with objects in one trial, the second state will be the stopping criterion.

**Action Space.**  The RL policy predicts two grippers' orientations $R_1, R_2 \in SO(3)$, which is the 3D rotation group about the origin of three-dimensional Euclidean space under the operation of composition. Since a 6-dimensional real vector is enough to restore the $3 \times 3$ orientation matrix $R$, the dimension of the vector proposed by RL policy can be reduced to 12. Moreover, the actual actions of grippers are defined as: first generating two grippers at a certain distance $d_g$ away from contact points in the gripper fingers' orientation, then synchronously moving forward to the contact points, closing fingers, and finally moving back to the original positions.

**Reward Design.**  The reward is the summation of: 1) task reward, which is assigned 1 if the interaction is successful(as defined in B). If the object keeps steady (but its movement $d$ is less than the threshold), we give the reward of $20d$, so the longer it moves, the higher reward it will get. If the movement is over the threshold(but the object is rotated or toppled), it will acquire a small reward of 0.15. 2) interaction reward, which is determined by whether the object is grasped during the whole interaction. Successful grasps at the beginning and in the end will respectively give a reward of 0.5.

### E.2  IMPLEMENTATION AND TRAINING DETAILS

We leverage SAC (Haarnoja et al., 2018) to train the RL policy. It consists of a policy network and two soft-Q networks, all of which are implemented as Multi-layer Perceptions (MLP). The policy network receives the state as input ($\mathbb{R}^{13}$), and predicts the grippers' orientations as action ($\mathbb{R}^{12}$). Its network architecture is implemented as a 4-layer MLP ($13 \rightarrow 512 \rightarrow 512 \rightarrow 512 \rightarrow 512$), followed by two separate fully connected layers ($512 \rightarrow 12$) that estimate the mean and variance of action probabilities, respectively. The Q-value network receives both the state and action as input ($\mathbb{R}^{25}$), and predicts a single Q value ($\mathbb{R}^1$). Its network architecture is implemented as a 4-layer MLP ($25 \rightarrow 512 \rightarrow 512 \rightarrow 512 \rightarrow 1$).

We use the following hyper-parameters to train the RL policy: 16384 (buffer size); 512 (batch size); 0.0002 (learning rate of the Adam optimizer (Kingma & Ba, 2014) for both the policy and Q-value network);

## F    More Discussions on Collaborative Adaptation Procedure

In this section, we will provide more discussion on why the Collaborative Adaptation procedure can further improve the performance.

Before the Collaborative Adaptation procedure, we only use the collected offline data to train the networks. After such training stage, the proposed affordance maps and actions may not be completely and perfectly consistent with the desired distributions of collected offline data, and thus will exist some errors. The errors of the two separately trained modules would accumulate and lead to even worse performance, and thus chances are that the dual-gripper manipulation system will propose many actions (i.e., false positive actions) that cannot actually fulfill the tasks.

Therefore, we propose the Collaborative Adaptation procedure, let two grippers perform actions end-to-end in simulation, and use the actual action results to adapt the dual-gripper manipulation system with the above explained cumulative errors. During this procedure, those false positive cases would be tuned to be negative, and thus the system will be adapted into a better collaborative system.

## G    Computational Costs, Timing and Error Bars

It takes around 4.5-6 days per experiment on a single Nvidia 3090 GPU. Specifically, it takes around 3.5 days and 1 day to respectively train the networks in Perception Module and in the subsequent Collaborative Adaptation procedure, while an extra 1.5 days is needed if we apply the RL Augmented Data Sampling.

Our model only consumes 1,600 MB memory of the GPU during the inference time.

For each task, for the reason that we randomly sample 500 seeds initializing 500 random configurations, representing 500 different targets and object states, and we execute 3 trials for each configuration to compute the average success rate, the errors may be quite small.

## H    More Details about Baselines

### H.1    Heuristic Baseline

For the rule-based heuristic baseline, we hand-craft a set of rules for different tasks. We now describe the details in the following:

- Pushing: to choose where to manipulate, we select two contact points $p_1$ and $p_2$ that are halfway up the given object, where $p_1$ is at the left one-fifth of the length of the object, and $p_2$ is at the right one-fifth of the length of the object. Then, we initialize the two grippers with the same orientation as the given task, which is a unit vector denoting the object's goal pushing direction.

- Rotating: if the given task is to rotate an object clockwise on the horizontal plane, we will choose the first contact point $p_1$ at the one-tenth of the length at the right top, setting the first orientation downward, and thus the point is held as a fulcrum for rotation. Then we choose the second point $p_2$ halfway up the object, the greater the given rotation angle is, the closer it will be to the fulcrum, and the second orientation is set horizontally forward. On the contrary, if the given task is to rotate an object anticlockwise, the fulcrum will be chosen at the one-tenth of the length and at the left top.

- Toppling: we select two contact points $p_1$ and $p_2$ that are one-fifth up the given object, where $p_1$ is at the left fifth of the length of the object, and $p_2$ is at the right fifth of the length of the object. Then, we set the two grippers with the same orientation as the given task.

- Picking-up: we select the two contact points $p_1$ and $p_2$ on the objects' left and right top edges and set the two gripper orientations the same as the given picking-up direction.

### H.2    M-Where2Act Baseline

Since Where2Act (Mo et al., 2021) is used for single-gripper and task-less interactions, we should adapt it as a baseline by modifying its network structure.

For its Critic network, it receives as input the observation $O$, the given task $l$, the two contact points $p_1$, $p_2$, and the two orientations of the two grippers $R_1$, $R_2$ as a whole, and outputs a score $c \in [0, 1]$, indicating the success likelihood of accomplishing the task when the two grippers collaboratively manipulate this object.

To implement the Proposal network, we employ a conditional variational autoencoder (cVAE) (Sohn et al., 2015) instead of using a single MLP used in Where2Act. This module receives as input the observation $O$, the given task $l$, and both of the grippers' actions $u_1 = (p_1, R_1)$, $u_2 = (p_2, R_2)$. It proposes both gripper's orientations $R_1$, $R_2$ as a combination.

For its Affordance network, it receives as input the observation $O$, the given task $l$, and the two contact points $p_1$, $p_2$, and outputs a score $a \in [0, 1]$, indicating the success likelihood when grippers manipulate with this pair of contact points. The ground truth supervision of this module is provided by the modified Critic network and the modified Proposal network.

The loss functions are similar to our framework.

### H.3    M-FABRICFLOWNET BASELINE

We additionally compare our method with another baseline method named M-FabricFlowNet (M-FFN). We will first give a brief introduction to FabricFlowNet (Weng et al., 2022), and then provide the experimental results on different tasks.

FabricFlowNet uses two grippers to manipulate the cloth with a goal-conditioned flow-based policy. Given the initial and the target observations, FabricFlowNet can (1) use the FlowNet to predict the 2D optical flow, which represents the correspondence between two observations, (2) use the PickNet to take in the flow as input and predict the two pick points, (3) use the predicted flow at the pick points to find the two corresponding place points. Note that if the distance between pick points (or place points) is smaller than a threshold, they will use only a single gripper for manipulation to avoid the collision.

In our implementation, we adapt FabricFlowNet as a baseline with some modifications: (1) for optical flow, we calculate the ground-truth 3D flow between the initial observation and the target observation, and the ground-truth flow will be directly fed to the following PickNet. This means, we have the perfect flow. (2) For the PickNet, given the ground-truth flow, we adapt it from a 2D-version to a 3D-version by changing the backbone from FCN to PointNet++, and keeping other settings the same. (3) For the placing position, we also use the flow direction at the pick point to find the place point. However, since our task is in 3D space, apart from the place position, we also require to set the two grippers' orientations. For fair comparison, we use our pretrained Actor Networks to propose the two grippers' orientations.

Table 5 presents the sample-success-rate, and we can see that our method outperforms M-FFN. The reason is that, M-FFN only disentangles the learning of the two grippers' actions, without further designs for the collaboration. M-FFN only learns the second action conditioned on the first, while our method makes both the two actions conditioned on each other. Specifically, M-FFN uses ground-truth data to directly supervise the first action, and then uses ground-truth data to supervise the second action conditioned on the first. As for our method, we also use ground-truth data to supervise the second action conditioned on the first, but further, we use the previously trained second gripper module to supervise the first action conditioned on the second (as shown in Figure 4 in our main paper). Therefore, our method makes the two gripper modules aware of each other, while in M-FFN, only the second action module is aware of the first.

Table 5: Baseline comparison on the sample-success-rate metric.

| | Train Categories | | | | Test Categories | | | |
|---|---|---|---|---|---|---|---|---|
| | pushing | rotating | toppling | picking-up | pushing | rotating | toppling | picking-up |
| M-FFN | 33.67 | 19.60 | 47.20 | 19.30 | 25.40 | 14.80 | 36.47 | 26.90 |
| Ours | **48.76** | **33.73** | **65.53** | **40.33** | **42.93** | **35.07** | **41.80** | **54.33** |

# I    MORE RESULTS AND ANALYSIS

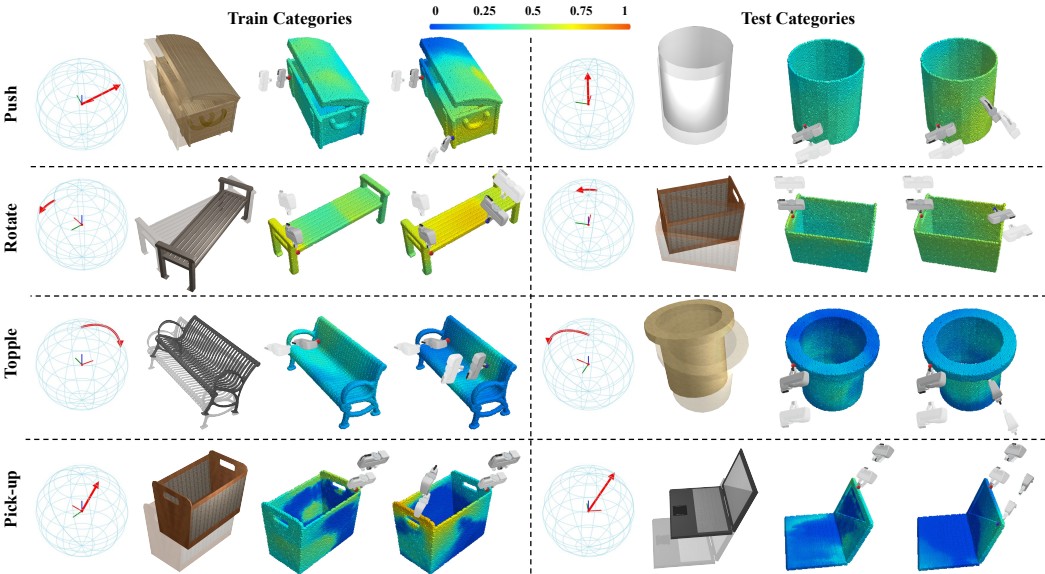

Figure 10: We present additional qualitative results of the learned Affordance networks $\mathscr{A}_1$ and $\mathscr{A}_2$ to augment Fig.4 in the main paper. In each block, from left to right, we respectively show the task represented by a red arrow, the object which needs to be moved from transparent to solid, the actionability heatmap predicted by network $\mathscr{A}_1$, and the actionability heatmap predicted by network $\mathscr{A}_2$ conditioned on the action of the first gripper.

## I.1    MORE QUALITATIVE RESULTS

In Figure 10 and 11, we respectively show more results on the learned Affordance networks $\mathscr{A}$ and Critic networks $\mathscr{C}$.

## I.2    FAILURE CASES

Figure 12 visualizes some failure cases, which demonstrate the difficulty of our tasks and some ambiguous cases that are hard for robots to figure out. See the caption for detailed descriptions.

# J    DISCUSSION OF LIMITATIONS

Though our work takes one important step toward the dual-gripper collaborative affordance prediction, it focuses on the short-term interaction while there exist many long-term tasks in our daily life, such as lifting and moving a heavy pot to a target pose with two grippers, opening a drawer with one gripper and putting an object inside with the other gripper. These long-term tasks are much more difficult and challenging, and we leave this topic for future work.

A limitation of our work is that we use the flying grippers instead of the robot arms to interact with objects. Following Where2Act (Mo et al., 2021), we abstract away the control and planning complexity of having robot arms, and focus on learning the collaborative affordance for dual-gripper manipulation tasks. Though our real-robot experiments show that the proposed actions by our network can apply to real robot arms with the help of the motion planning in MoveIt! (Chitta et al., 2012), it is meaningful and more realistic to use the robot arms. Because, in the real-robot experiments, we should additionally consider the degree limitations of each arm joint, which means some contact points may be out of reach. Moreover, we must take into consideration the collisions amongst the joints of different robots to ensure safety, which will be more challenging compared to simply avoiding the collisions between two flying grippers.

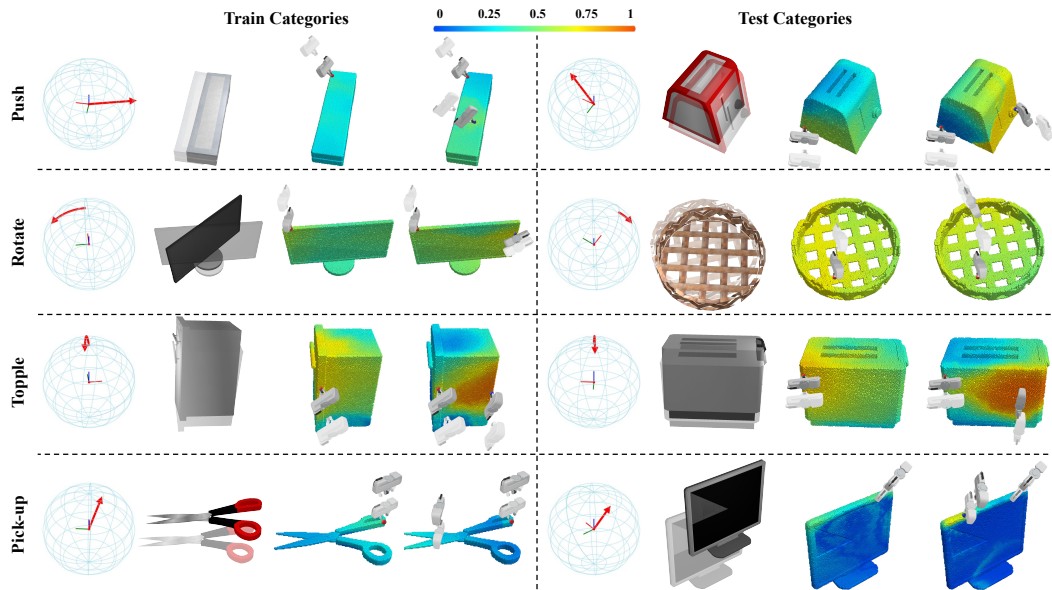

Figure 11: We visualize additional qualitative results of the per-point action scores predicted by our Critic networks $\mathscr{C}_1$ and $\mathscr{C}_2$ to augment Fig.5 in the main paper. In each result block, from left to right, we show the task represented by a red arrow, the input shape, $\mathscr{C}_1$'s predictions of success likelihood applying the first action over all points, and $\mathscr{C}_2$'s predictions of success likelihood applying the second action over all points given the first action.

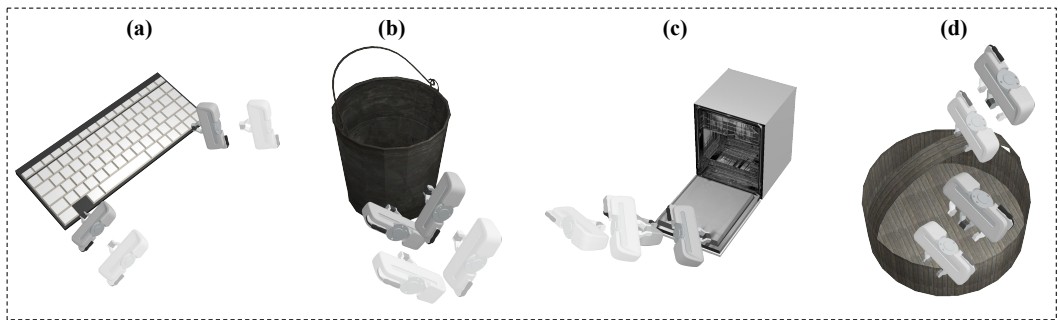

Figure 12: We visualize some failure cases, which demonstrate the difficulty of the tasks and some ambiguous cases that are hard for robots to figure out. (a) To push a keyboard on the ground, the two grippers may have collisions with the ground and can not move on. (b) To topple a cylinder like a bucket, it is easy to make the cylinder rotate during toppling. (c) The door is open due to gravity, so we can hardly find suitable contact points for interaction. (d) A failed picking-up attempt since the lifting handle is a little oversized for the gripper to grasp.

