# OpenReview forum: "DualAfford: Learning Collaborative Visual Affordance for Dual-gripper Manipulation"
_ICLR.cc/2023/Conference — ICLR 2023 poster_

### Official Review · Reviewer_VYuj · 2022-10-22

**Confidence:** 4
**Clarity, Quality, Novelty And Reproducibility:** 1. The quality writing is great and e…
**Correctness:** 3
**Technical Novelty And Significance:** 2
**Empirical Novelty And Significance:** 2
**Recommendation:** 8

**Strength And Weaknesses:**

Strength:
1. The authors tackle a challenging problem, i.e., dual-gripper manipulation. The work is timely toward home-assistive robots.
2. . The reviewer believes the work can motivate the community of visual affordance learning and dual-gripper manipulation to stimulate new ideas toward home-assistive applications.
3. The authors propose a collaborative visual affordance learning framework to address the challenges of dual-gripper manipulation due to high-degree freedoms. Specifically, they propose to disentangle the dual-gripper learning problem into two separate yet coupled subtasks, i.e., the second gripper module is conditioned on the output of the first module. With the proposed design, it can reduce the search space. In addition, they propose a collaborative adaptation procedure to enhance cooperation. The proposed collaborative adaptation procedure is shown to be effective in Table 1, where the success rate on both the training and testing category is higher than the one architecture without a collaborative adaptation procedure.
4. The authors demonstrate the feasibility of the proposed method on real robots that demonstrate dual-gripper manipulation on the following tasks, i.e., picking up containers, rotating boxes, pushing displays, and toppling buckets.

Weaknesses:
1. An essential piece of this work is the concept of collaborative visual affordance. However, the authors did not propose any metrics and benchmarks for this part. The reviewer found it critical as this is the "direction" the authors aim to promote. Otherwise, the authors should compare other bimanual manipulation approaches (e.g., Weng et al., 2021 and Xie et al., 2020)
2. If visual affordance learning "direction" is the aim of this work, what are the limitations of other visual actionable approaches for the tasks tackled in this work? For instance, Weng et al. propose a flow-based method for bimanual cloth manipulation (Weng et al., FabricFlowNet: Bimanual Cloth Manipulation with a Flow-based Policy, CoRL 2021). The work shows the feasibility of bimanual cloth manipulation. While the task domain of Weng et al., 2021 is different from this work, the idea of disentangling has been discussed in Figure 3. In addition, the pick point prediction is another strategy for manipulation. What is the advantage of the proposed approach? The authors should discuss the work and the difference.
3. One of the contributions is the benchmark based on SAPIEN. However, the authors did not discuss the "value" of the proposed benchmark over others (e.g., Xie et al., 2020, Chitnis et al., 2020, Chen et al., 2022). Please comment.
4. Lack of detailed ablative studies: This work has several components for collaborative visual affordance learning. Specifically,
    1. Disentangled gripper module design with conditioning. What is the performance without conditioning? Does the model fail to converge?
    2. Collaborative adaptation procedure
    3. RL data sampling vs. Random sampling
5. Currently, only (b) is evaluated and shown its effectiveness. The other two aspects do not prove their values. The question regards (c) can be found in the next point. Please comment.
6. Table 2 results are confusing: Specifically, based on the reviewer's understanding, the results aim to show that the success rate of the RL-based sampling method is more effective than a random method. However, the notion in Table 2 is "Ours w/o RL" and "Ours," which is confusing. Moreover, the results are for the picking-up task. How about other tasks? Please comment.
7. Diversity of data collected using RL policy: while the authors show the success rate of RL-based sampling in Table 2, the reviewer is interested in the diversity of the RL-based sampling. What is the coverage of the sampled points based on the RL-based method? Could we visualize them?
8. The reviewer suggests that the authors can conduct experiments in the real world where a policy is trained on training categories but test on the testing categories to further strengthen the proposed method's value.

**Summary Of The Paper:**

It is known that dual gripper is a challenging yet valuable robotic problem because there are daily tasks that are hard to complete with a single gripper. When robots are deployed for home-assistive applications, it becomes even more difficult because of the diverse 3D objects in the daily environment. The authors propose a collaborative visual affordance to facilitate dual-gripper manipulation to overcome the hurdle. Dual-gripper manipulation has the problem of a high degree of freedom. The authors propose a learning framework that disentangles the two gripper pose prediction subtasks and optimizes the two subtasks via a collaborative adaptation procedure. The authors conduct experiments on a proposed benchmark based on the PartNet-Mobility and ShareNet datasets. Four diverse dual-gripper manipulation tasks, i.e., pushing, rotating, toppling, and picking up, are studied. The proposed method achieves good performance compared with three baselines (random approach, heuristic approach, and a modification to Where2Act [Mo et al., 2021]). Additional experiments involve (1) showing the effectiveness of an RL-based data collection strategy and (2) real-world robot demonstrations.

**Summary Of The Review:**

The authors propose a collaborative visual affordance learning framework for dual-gripper manipulation. In addition, a benchmark is proposed to validate the effectiveness of different algorithms for dual-gripper manipulation. In addition, a real-world demonstration is presented to prove the effectiveness of the proposed approach. However, several concerns are raised in the Weakness section, including the novelty of the proposed framework and benchmark and insufficient ablative studies.

---

> ### Author Response · Authors · 2022-11-16
> **REPLY: Thank you for your valuable suggestions! We have addressed your concerns and revised our paper accordingly. Hope to hear back from you if you have further questions. [Part 5/5]**
>
> Thank you again for your valuable suggestions and questions! In summary, our paper proposes a novel framework, DualAfford. It focuses on studying the **collaborative visual affordance** for dual-gripper manipulation, not only disentangles the learning of the two grippers' actions, making the second gripper module aware of the first, but also uses the trained second gripper module to supervise the first action, making the first gripper module aware of the second (as shown in our paper's Figure 4). Besides, we also add more experimental results including the **FabricFlowNet baseline**, the sampling efficiency comparison, the ablation study, and the visualization of the RL sampled data. We also give more discussion in the above response, and we sincerely hope that our response makes things clearer to you and addresses your concerns well.  Please feel free to let us know if you have further questions, and we are happy to take further discussion!
>
>
> * [Paper-1] Thomas Weng, Sujay Man Bajracharya, Yufei Wang, Khush Agrawal, and David Held. Fabricflownet: Bimanual cloth manipulation with a flow-based policy. In Conference on Robot Learning, pp. 192–202. PMLR, 2022.
>
> * [Paper-2] Zeng, A., Song, S., Welker, S., Lee, J., Rodriguez, A., and Funkhouser, T. (2018). Learning synergies between pushing and grasping with self-supervised deep reinforcement learning. In 2018 IEEE/RSJ International Conference on Intelligent Robots and Systems (IROS), pages 4238–4245. IEEE. Best Cognitive Robotics Paper Award Finalist, IROS
>
> * [Paper-3] Kaichun Mo, Leonidas J. Guibas, Mustafa Mukadam, Abhinav Gupta, and Shubham Tulsiani. Where2act: From pixels to actions for articulated 3d objects. In Proceedings of the IEEE/CVF International Conference on Computer Vision (ICCV), pp. 6813–6823, October 2021.
>
> * [Paper-4] Samir Yitzhak Gadre, Kiana Ehsani, and Shuran Song. Act the part: Learning interaction strategies for articulated object part discovery. In Proceedings of the IEEE/CVF International Conference on Computer Vision (ICCV), pp. 15752–15761, October 2021.
>
> * [Paper-5] Ruihai Wu, Yan Zhao, Kaichun Mo, Zizheng Guo, Yian Wang, Tianhao Wu, Qingnan Fan, Xuelin Chen, Leonidas Guibas, and Hao Dong. VAT-mart: Learning visual action trajectory proposals for manipulating 3d ARTiculated objects. In International Conference on Learning Representations, 2022.
>
> * [Paper-6] K. Zakka, A. Zeng, J. Lee, and S. Song. Form2fit: Learning shape priors for generalizable assembly from disassembly. IEEE International Conference on Robotics and Automation (ICRA), 2020.
>
> * [Paper-7] Fan Xie, Alexander Chowdhury, M De Paolis Kaluza, Linfeng Zhao, Lawson Wong, and Rose Yu. Deep imitation learning for bimanual robotic manipulation. Advances in neural information processing systems, 33:2327–2337, 2020.
>
> * [Paper-8] Kaichun Mo, Shilin Zhu, Angel X Chang, Li Yi, Subarna Tripathi, Leonidas J Guibas, and Hao Su. Partnet: A large-scale benchmark for fine-grained and hierarchical part-level 3d object understanding. In Proceedings of the IEEE/CVF Conference on Computer Vision and Pattern Recognition, pp. 909–918, 2019.
>
> * [Paper-9] Fanbo Xiang, Yuzhe Qin, Kaichun Mo, Yikuan Xia, Hao Zhu, Fangchen Liu, Minghua Liu, Hanxiao Jiang, Yifu Yuan, He Wang, et al. Sapien: A simulated part-based interactive environment. In Proceedings of the IEEE/CVF Conference on Computer Vision and Pattern Recognition, pp. 11097–11107, 2020.
>
> * [Paper-10] Angel X Chang, Thomas Funkhouser, Leonidas Guibas, Pat Hanrahan, Qixing Huang, Zimo Li, Silvio Savarese, Manolis Savva, Shuran Song, Hao Su, et al. Shapenet: An information-rich 3d model repository. arXiv preprint arXiv:1512.03012, 2015.
>
> * [Paper-11] Rohan Chitnis, Shubham Tulsiani, Saurabh Gupta, and Abhinav Gupta. Efficient bimanual manip- ulation using learned task schemas. In 2020 IEEE International Conference on Robotics and Automation (ICRA), pp. 1149–1155. IEEE, 2020.
>
> * [Paper-12] Yuanpei Chen, Yaodong Yang, Tianhao Wu, Shengjie Wang, Xidong Feng, Jiechuang Jiang, Stephen Marcus McAleer, Hao Dong, Zongqing Lu, and Song-Chun Zhu. Towards human-level bimanual dexterous manipulation with reinforcement learning, 2022.

---

> ### Author Response · Authors · 2022-11-16
> **REPLY: Thank you for your valuable suggestions! We have addressed your concerns and revised our paper accordingly. Hope to hear back from you if you have further questions. [Part 4/5]**
>
> > The notion in Table 2 is "Ours w/o RL" and "Ours," which is confusing. Moreover, the results are for the picking-up task. How about other tasks?
>
> We are sorry for this confusion. Table 2 aims to show the success rate of the random sampling method and the RL-based sampling method, and we have revised the notion to **"Random-Sampling"** and **"RL-Sampling"**.
> In our work, following Where2Act [Paper-2], we first use the random sampling method to collect offline data. We find that in the pushing, rotating, and toppling tasks, we are able to collect sufficient data using the random sampling method as the success rate is larger than 1%. Therefore, we only use the random sampling method for these three tasks, since the data collection method is not our main contribution. However, for the picking-up task, we find it is nearly impossible to collect positive data because the success rate is only around 0.01%. To deal with this problem and to efficiently collect sufficient data, we then train an RL-based sampling method for the picking-up task. Therefore, the RL sampling method is optional, depending on the efficiency of the random sampling method.
>
> During rebuttal, we also train the RL method on the pushing task, and we test the data sampling success rate of the random sampling method and the RL-based sampling method. Here are the results:
>
> **Random-Sampling**: Box 3.00%, Dishwasher 0.05%, Display 1.90%, Bench 2.40%, Keyboard 8.8%, Kitchen Pot 2.00%, Toaster 1.40%, Basket 1.10%.
>
> **RL-Sampling**: Box 20.18%, Dishwasher 1.45%, Display 6.62%, Bench 6.50%, Keyboard 15.30%, Kitchen Pot 6.40%, Toaster 8.00%, Basket 3.35%.
>
> Due to the limitation of time during rebuttal, we only train and test the RL policy on the pushing task, and we are willing to train the RL policy on other tasks and present the results in the final paper.
>
>
> > detailed ablative studies (3). RL data sampling vs. Random sampling
>
> Thank you for this suggestion. As we explained in the previous question, due to time limitations, we use the pushing task as an example to train the RL sampling policy, and we train our model on the pushing task with data collected by RL. With **RL sampled data**, it receives success rates of **41.53% and 37.93%** respectively on the training categories and the test categories. With **Randomly sampled data**, it receives success rates of **48.76% and 42.93%** respectively. The policy trained on randomly sampled data has better performace, because its training data are more diverse, so it can better generalize to novel shapes and novel categories. We are willing to further do this ablation study on the rotating and toppling tasks. However, for the picking-up task, since it is nearly impossible to collect successful interaction data using the random-sampling method, this ablation study can not be conducted on the picking-up task.
>
> > Diversity of data collected using RL policy. Could we visualize them?
>
> We visualize some data collected by our RL policy at the link [https://sites.google.com/view/dualafford-rebuttal](https://sites.google.com/view/dualafford-rebuttal) We can see that, for the same object instance, the RL policy can collect diverse data with different contact points and orientations.
>
> > The reviewer suggests that the authors can conduct experiments in the real world where a policy is trained on training categories but test on the testing categories to further strengthen the proposed method's value.
>
> Thank you for this suggestion. We agree that training our policy in the real world instead of in the physics simulation is meaningful. However, we are sorry that it is hard to be accomplished during the rebuttal period, since it requires hundreds of objects with thousands of diverse demonstrations to train our framework, while the data is difficult to collect in the real world. As we have demonstrated that our framework can work in some real-world objects without any finetuning on them, we believe with enough and diverse real-world data, our framework will achieve good results on the experiment you point out.

---

> ### Author Response · Authors · 2022-11-16
> **REPLY: Thank you for your valuable suggestions! We have addressed your concerns and revised our paper accordingly. Hope to hear back from you if you have further questions. [Part 3/5]**
>
> **For affordance learning**, we discuss the advantage of our affordance learning approach compared to other strategies. (1) **FarbicFlowNet** (Weng et al., 2021) [Paper-1]. For each data point, FabricFlowNet uses the ground-truth pick point (with 2D Gaussian centered at the pick point) to supervise the proposed picking heatmap. The ground-truth heatmap only highlights the region around the pick point, but penalties all the remaining regions. As a result, as shown in this paper's Figure 3, only one corner of points in the predicted heatmap by FabricFlowNet is highlighted, with other points not highlighted. However, for a given task, there may exist many kinds of collaborative actions that could accomplish the task, and the heatmap should contain many and diverse highlighted points. In comparison, we can see that our proposed affordance map covers much more diverse solutions and highlights many different but possible points. We can propose diverse interactions with a same object, which is more reasonable and more generalizable. (2) **Act the Part** (Gadre et al., 2021) [Paper-2]. As we explained in the **previous question**, they use one gripper to hold one articulated part, and use the other gripper to move another part. They use two heatmaps to separately highlight where to hold or push. However, they aim to **segment articulated parts** from interactive perception without focusing on collaboration, but we propose a generic framework that enables the two grippers to collaboratively accomplish different kinds of tasks for the target object.
>
>
> > Discuss the "value" of the proposed benchmark over others (e.g., Xie et al., 2020, Chitnis et al., 2020, Chen et al., 2022).
>
> Thank you for this suggestion. Using the large-scale PartNet-Mobility [Paper 8-9] and ShapeNet [Paper-10] datasets, our benchmark contains much more object instances covering **different categories**, so the visual observations have **diverse geometries**. Our benchmark requires not only the collaboration between two grippers, but also the strong **generalization ability** to **novel unseen categories**. In comparison, both Xie et al., 2020 [Paper-7] and Chitnis et al., 2020 [Paper-11] have one object instance for each task, and thus their requirement on the model's generalization ability is limited. Besides, Chen et al., 2022 [Paper-12] use object states as input, while our benchmark uses object visual observations (point clouds) as input. It is more generic and real to use object visual observations as input, for the reason that it is hard for a robot to get object states, and object states could not fully represent the target object (e.g., states could not fully represent the geometry of the object).
>
> > detailed ablative studies (1). Disentangled gripper module design with conditioning. What is the performance without conditioning? Does the model fail to converge?
>
> Thank you for this suggestion. M-Where2Act (a dual-gripper Where2Act [Paper-2] baseline) is the method without conditioning, and we have compared with it in our paper. M-Where2Act is directly adapted from Where2Act, and considers the dual grippers as a combination. Concretely, its Affordance Network proposes the combination of two contact points, its Actor Network proposes the combination of two orientations, and its Critic scores the combination of two gripper actions. This model can converge but has worse performance, because it has to learn the intrinsically quadratic problem without disentangling. Taking the pushing task as an example, M-Where2Act receives an accuracy of **28.00%**, while our method receives an accuracy of **48.76%** (more results are in **Table 1** in our paper).

---

> ### Author Response · Authors · 2022-11-16
> **REPLY: Thank you for your valuable suggestions! We have addressed your concerns and revised our paper accordingly. Hope to hear back from you if you have further questions. [Part 2/5]**
>
> > If visual affordance learning "direction" is the aim of this work, what are the limitations of other visual actionable approaches for the tasks tackled in this work? For instance, Weng et al. propose a flow-based method for bimanual cloth manipulation. The work shows the feasibility of bimanual cloth manipulation. While the task domain of Weng et al., 2021 is different from this work, the idea of disentangling has been discussed in Figure 3. In addition, the pick point prediction is another strategy for manipulation. What is the advantage of the proposed approach? The authors should discuss the work and the difference.
>
> Thank you for this valuable question! In the following paragraphs, first, **for bimanual manipulation**, we explain the difference between our method and **FabricFlowNet**(Weng et al., 2021 [Paper-1]), which uses the **similar idea of disentangling** but takes **less consideration for the collaboration** between the two grippers, and we further use FabricFlowNet as a baseline and analyze the experimental results. Besides, **for affordance learning**, we discuss the advantage of our affordance learning approach compared to other strategies like the pick point prediction.
>
> **For bimanual manipulation, FabricFlowNet only disentangles** the learning of the two grippers' actions, but **without further designs for the collaboration.** FabricFlowNet only learns the second action conditioned on the first, while we make both the two actions conditioned on each other. Specifically, FabricFlowNet uses ground-truth data to directly supervise the first action, and then uses ground-truth data to supervise the second action conditioned on the first. As for our method, we also use ground-truth data to supervise the second action conditioned on the first, but further, we use the previously trained second gripper module to supervise the first action conditioned on the second (as shown in Figure 4 in our paper). Therefore, our method makes the two gripper modules aware of each other, while in FabricFlowNet, only the second action module is aware of the first.
>
> We **adapt FabricFlowNet as a baseline**. The original FabricFlowNet uses two grippers to manipulate the cloth to a goal pose with a flow-based policy. Given the initial and the target observations, FabricFlowNet (1) uses the FlowNet to predict the 2D optical flow, which represents the correspondence between two observations, (2) uses the PickNet to take in the flow as input and predict the two pick points, (3) use the predicted flow at the two pick points to find the two corresponding place points.
>
> In our implementation, (1) for optical flow, we calculate the ground-truth 3D flow between the initial observation and the target observation, and the ground-truth flow will be directly fed to the following PickNet. This means we have the perfect flow. (2) For the PickNet, given the ground-truth flow, we adapt it from a 2D-version to a 3D-version by changing the backbone from FCN to PointNet++, and keep other settings the same. (3) For the placing pose, since our task is in 3D space, apart from the place position, we also require the two grippers' movement direction. We design two methods to predict place pose: (i) using the flow direction at the pick point as the gripper's movement direction, and calculating the place position according to the flow; (ii) using our two Actor Networks to propose the two gripper's orientations, and calculating the place position according to the predicted orientations.
>
> For the **experimental results on the FabricFlowNet baseline**, on the pushing task, method (i) of the FabricFlowNet baseline achieves a success rate of **27.40%**; method (ii) achieves a success rate of **33.67%**. In comparison, our DualAfford's success rate is **48.76%**. (Due to the time limitation during rebuttal, we only train on the pushing task, and we are willing to add this baseline on other tasks in the final paper.) The reason why method (ii) outperforms method (i) is that, although the two grippers' movement directions are proposed the same as the object's movement flow, the movement direction of the object will not be the same as the two grippers' movement directions. Our method achieves the best result, because it lays more focus on the collaboration between the two grippers, as explained in the above paragraph.

---

> ### Author Response · Authors · 2022-11-16
> **Thank you for your valuable suggestions! We have addressed your concerns and revised our paper accordingly. Hope to hear back from you if you have further questions. [Part 1/5]**
>
> We sincerely thank you for taking the time and effort to review our paper. Thank you for the constructive feedback and valuable suggestions for further improving our work, and we have addressed all the questions below.
>
> > An essential piece of this work is the concept of collaborative visual affordance. However, the authors did not propose any metrics and benchmarks for this part. The reviewer found it critical as this is the "direction" the authors aim to promote. Otherwise, the authors should compare other bimanual manipulation approaches (e.g., Weng et al., 2021 and Xie et al., 2020).
>
> Thank you for this suggestion. First, we explain why we can use the success rate of downstream tasks as the evaluation metric of affordance. Specifically, we use success rate of different bimanual manipulation tasks to evaluate our proposed collaborative visual affordance. Then, we compare our method with other bimanual manipulation approaches you mentioned.
>
> For **the evaluation metric of collaborative visual affordance**, by definition, affordance indicates the opportunities of interaction, and thus affordance is highly related to the downstream tasks. Therefore, the success rate of downstream tasks is widely used as the evaluation metric of whether the learned affordance is good or not [Paper 1-6]. Besides, these works have not designed any additional metrics about affordance itself, as the metric of success rate well satisfies the conception of affordance. Similar to these works and following Where2Act [Paper-3], we use the metric sample-success-rate to evaluate the affordance. With high-quality affordance maps, the accuracy of downstream tasks will be improved.
>
> Also, we **compare our method with other bimanual manipulation approaches**:
>
> (1) **FabricFlowNet** (Weng et al., 2021) [Paper-1]. It is an important baseline of our method, we will discuss the method and its difference from ours, explain the experiment details, and analyze the results **in the next question**.
>
> (2) **Act the Part** (Gadre et al., 2021) [Paper-2]. Its task is to **segment articulated parts** of articulated objects (e.g. eyeglasses) from interactive perception. This method uses one gripper to hold one articulated part (e.g. one eyeglass leg), and uses the other gripper to move another part (e.g. another eyeglass leg). The observstions before and after the interaction will be used to segment articulated parts. However, in our paper, we propose a generic framework that enables the two grippers to collaboratively accomplish different tasks for the target object, such as pushing or picking up a heavy box. In conclusion, the tasks of the two works are totally different, our method is designed for collaboration, while Act the Part is designed for segmentation, so Act the Part is not suitable for our collaborative task.
>
> (3) **HDR-IL** (Xie et al., 2020) [Paper-7]. This paper designs a deep imitation learning framework for bimanual manipulation. This method requires extra human annotations, besides, it is trained and tested on one fixed object instance. In comparison, our method is self-supervised without any human annotations or demonstrations. Also, our method is trained on hundreds of objects covering different categories, and has the ability to generalize to novel unseen categories. The settings of the two works are quite different, so it is not suitable to use HDR-IL as our baseline.

---

> ### Author Response · Authors · 2022-11-19
> **Looking Forward to Seeing Your Response!**
>
> Dear reviewer VYuj,
>
> Thanks again for your valuable suggestions! Given the discussion phase is quickly passing, we want to know if our response resolves your concerns. If you have any further questions, we are more than happy to discuss them. We are looking forward to seeing your response!
>
> Best, All anonymous authors

---

> ### Author Response · Authors · 2022-12-07
> **Looking Forward to Seeing Your Reply!**
>
> Dear reviewer VYuj,
>
> As the rebuttal deadline is coming, we wonder if you have any further questions or suggestions about our response, and we are more than happy to discuss them. Thank again for your support of this work!
>
> Best, All anonymous authors.

---

### Official Review · Reviewer_yumu · 2022-10-22

**Confidence:** 4
**Correctness:** 4
**Technical Novelty And Significance:** 3
**Empirical Novelty And Significance:** 4
**Recommendation:** 8

**Clarity, Quality, Novelty And Reproducibility:**

- Very well organized and thorough paper: experiments are detailed, and include real-world evaluation, which is a very significant bar for robotics papers seldom met in practice, but essential to deciding on the value of any approach that pertains to real-world robotics.
- Good limitations section.
- Great video


**Strength And Weaknesses:**

Strengths:
- very important topic, extremely relevant to representation learning for robotics. While it's been relatively easy to learn single-arm grasping directly from manipulation data, without having to learn an explicit affordance representation, doing so in the context of leads to an explosion in complexity which makes learning an independent proposal model for affordances very attractive.
- the model appears sound, well-motivated, and immediately useful to derive further research on the topic.
- the approach to combating the combinatorial explosion of having to learn two joint affordance models together is well-motivated, clever, and of practical relevance to any setting where chained conditional models have to be learned efficiently.

Weaknesses:
- the main weakness of the approach, in that it doesn't take the robot kinematics into account, is well documented and not a showstopper for future work.
- the benchmark developed as part of this work would be exceedingly valuable to the community, particularly since the success rate on this task seems to hover around ~50%, which indicates it's in the sweet spot of not being too difficult or too easy. I saw no discussion of providing it as an open-source benchmark, which would greatly enhance the value of this submission.

**Summary Of The Paper:**

This paper proposes an approach to learn a bimanual perceptual affordance model, provides a simulated benchmark to evaluate the model and compares its performance against a number of sensible baselines.

**Summary Of The Review:**

Very nice paper overall, useful to the community with a few ideas that could apply beyond the narrow setting of affordance modeling.

---

> ### Author Response · Authors · 2022-11-16
> **REPLY: Thank you for your positive feedback! We have addressed your concerns and revised our paper accordingly. Hope to hear back from you if you have further questions.**
>
> We sincerely thank you for your positive feedback and helpful suggestions for further improving our work.
> It is our great pleasure to hear that you recognize the contribution of our work. We have addressed all the questions below and revised the paper accordingly to reflect our changes (we highlight all changes in Red in the revised paper).
>
> > the main weakness of the approach, in that it doesn't take the robot kinematics into account, is well documented and not a showstopper for future work.
>
> Thank you for this suggestion. We agree that the absence of robot kinematics is not a showstopper for future work but a limitation of our work. We have revised Section I of Supplementary (Discussion of Limitations) to make it more clear, as follows:
>
> A limitation of our work is that we use the flying grippers instead of the robot arms to interact with objects. Following Where2Act (Mo et al., 2021), we abstract away the control and planning complexity of having robot arms, and focus on learning the collaborative affordance for dual-gripper manipulation tasks. Though our real-robot experiments show that the proposed actions by our network can apply to real robot arms with the help of the motion planning in MoveIt! (Chitta et al., 2012), it is meaningful and more realistic to use the robot arms. ...
>
> > the benchmark developed as part of this work would be exceedingly valuable to the community, ... no discussion of providing it as an open-source benchmark
>
> Thank you for your appreciation! We will open-source our benchmark (including all of the code and dataset) upon acceptance. We will provide detailed information on this benchmark such as environment specifics, task design, data collection, and evaluation configurations. We will also provide the pretrained DualAfford model, as well as the training code. We also hope that our benchmark will be valuable to the community and facilitate future research on dual-gripper manipulation.
>
> We sincerely hope that our response above can address your concerns well. Please feel free to let us know if you have further questions. Thank you again for your time and your valuable comments!

---

> ### Author Response · Authors · 2022-11-19
> **Looking Forward to Seeing Your Response!**
>
> Dear reviewer yumu,
>
> As the discussion phase is quickly passing, we want to know if you have any further questions or suggestions, and we are more than happy to discuss. Thanks again for your valuable reviews!
>
> Best, All anonymous authors

---

> > ### Comment · Reviewer_yumu · 2022-11-21
> > **Thank you.**
> >
> > Nothing to add. Noted that authors intend to open source the benchmark, which is great!

---

### Official Review · Reviewer_RF6q · 2022-11-01

**Confidence:** 3
**Correctness:** 3
**Technical Novelty And Significance:** 2
**Empirical Novelty And Significance:** 3
**Recommendation:** 6

**Clarity, Quality, Novelty And Reproducibility:**

Clarity. The paper is overall well written The messages are clear and follow a logical progression throughout the text.

Novelty: The authors propose an original work on extending visual actionable affordance detection for dual-gripper collaborative manipulation. This work innovates in vision-based dual-arm collaborative manipulation which has immediate applications in service robotics.

Technical Quality: Technical details are presented and seem generally sound, there are a few parts that would require additional analysis.
Clarity: The paper is overall well written The messages are clear and follow a logical progression throughout the text.

Reproducibility The authors provide details about hyper-parameter and architecture configurations, as well as a detailed setup of evaluation metrics for the concerned tasks.

**Strength And Weaknesses:**

Strengths:
1. The paper is well-written, motivates its purposes, and presents its ideas in a clear fashion.
2. The proposed method addresses the complexity bottleneck of dual-arm manipulation by sequencing predictions from two grippers and conditioning the latter on the first.
3. The paper also proposes (and experimentally evaluates) an RL-based method for data generation.
4. Evaluation includes a study of generalization properties (novel shapes to be manipulated), which greatly improves its experimental
5. The proposed method is integrated with a real robot and includes demonstrations in real environments.
6. The authors generate evaluation testbeds in a simulation environment and claim that will release the dataset, which is a great contribution given the sparsity of resources for dual-arm manipulation.
Weaknesses
1. I would like more discussion on Sec. 4.4 (Collaborative Adaption Procedure), as it was also shown experimentally that it is essential for achieving collaboration between the grippers. However, the process is not 100% clear to me. It seems that the authors pair the learning networks with the robot to perform actions end-to-end in simulation, but how/why does that contribute more than training the components separately in an offline fashion, given that ground truth offline data are generated with annotations for the desired collaboration?
2. Evaluation baselines include a random policy, heuristics, and a dual-gripper-enhanced version of Where2Act (Mo et. al. 2021). I think the paper would benefit by comparing their method also with non-visual affordance-based methods for dual-arm manipulation. For example, how does this method compare with the mentioned related work keeping one gripper fixed method (Gadre et. al. 2021)?

**Summary Of The Paper:**

This paper extends the visual actionable affordance learning method of Where2Act (Mo et. al. 2021) for dual-gripper manipulation. To deal with the quadratic complexity of dual action space, the authors propose to sequence action predictions from two grippers and conditioning the second on the first. This reduces the action space to linear complexity and aids in ensuring collaborative policies of the two grippers. To train the network components of their architecture, the authors generate in offline fashion dense affordance and gripper orientation maps from a simulation environment by sampling random policies, as well as with a soft actor-critic RL method. The paper also proposes a collaborative adaptation procedure to enhance collaboration between the predicted actions by training the network components end-to-end. Experiment evaluation showcases that the proposed method outperforms task-specific heuristics and a quadratic complexity extension of Where2Act, while being able to generalize in unseen object geometries.

**Summary Of The Review:**

The paper introduces a novel method for dual-gripper manipulation based on visual affordances and it also publishes an evaluation suite for four collaborative tasks. I am fairly confident that the contributions are novel, impactful, and well-supported by the experimental section. However, I believe that the paper would benefit from some extra analysis and discussions on some aspects (see issues below):

1. I would suggest adding more discussion in Sec. 4.4 (see also weaknesses). The authors mention: ” [...] In this way, the two gripper modules can better understand whether their proposed actions are successful or not as they are aware of interaction results, and thus the two separately trained modules are integrated into one collaborative system”. How is this ”better understanding” achieved? How does training together add to a better understanding of collaboration, given that separate gripper-wise annotations are also collaboration-aware?

2. Some type of experimental confirmation of utilizing a dual-arm system vs a single gripper is missing. The authors motivate this well through the nature of the concerned task, but I think a single gripper could also achieve a subset of such tasks with extra effort (more actions). The benefits of then using a dual-gripper system are obvious, however, it would be nice if a comparison is included in the experimental section.

3. Comparison with other recent dual-gripper manipulation systems (e.g. Gadre et. al. 2021).

4. I would like to see a comparative analysis of the proposed vs. quadratic complexity version of Where2Act by training data size. How fast does your method pick up on the collaborative aspect compared to a holistic version?

5. There are a few typos, e.g. in the Conclusion section ”[...] its superiority of the three baselines” should be ”over” etc. I believe an extra iteration by the authors would resolve these issues.

---

> ### Author Response · Authors · 2022-11-16
> **REPLY: Thank you! We have addressed your concerns and revised our paper accordingly. Hope to hear back from you if you have further questions. [Part 2/2]**
>
> For the experimental results on FabricFlowNet baseline, on the pushing task, method (i) of the FabricFlowNet baseline achieves a success rate of **27.40%**; method (ii) achieves a success rate of **33.67%**. In comparison, our DualAfford's success rate is **48.76%**. (Due to the time limitation during rebuttal, we only train on the pushing task, and we are willing to add this baseline on other tasks in the final paper.) The reason why method (ii) outperforms method (i) is that, although the two grippers' movement directions are proposed the same as the object's movement flow, the movement direction of the object will not be the same as the two grippers' movement directions. Our method achieves the best result, because it lays more focus on the collaboration between the two grippers, as we explained in the above paragraph.
>
>
> > Some type of experimental confirmation of utilizing a dual-arm system vs a single gripper is missing.
>
> Thank you for this valuable suggestion!
>
> We fully agree that, although there are many tasks that one gripper cannot accomplish, a single gripper could also achieve a subset of such tasks with extra effort (more actions). Due to the limitation of time and computing resources during the rebuttal period, we take the pushing task as an example and conduct experiments to evaluate whether one gripper can work well in this kind of task.
> We allow the single gripper to operate for up to 5 steps. At each step, the input is the initial observation, the current observation, and the given task. The network architecture of this single-gripper version is similar to our dual-gripper version, except that it only needs to predict a single action. The accuracy of the single-gripper version is **20.60%**, while the accuracy of the dual-gripper version is **48.76%**.
>
> Experimental results show that, a single gripper could also achieve a subset of such tasks with extra effort (more actions), but single gripper's performance is worse than our dual-gripper framework. We find that there exist many situations that only one gripper cannot handle. For example, to push a display, using a single gripper is likely to make the display rotate, so the trained policy tends to push the display on the left side at the first step, and push on the right side at the second step... during the 5-step interaction, it is not easy to make the display regain the original orientation and to success. Besides, for a heavy dishwasher, it is hard for a single gripper to move it. Therefore, the performance of the single-gripper version is worse than the dual-gripper version.
>
> Thanks again for your suggestion! We are willing to add this experiment and analysis on all tasks in the final paper.
>
> > A comparative analysis of the proposed vs. quadratic complexity version of Where2Act by training data size. How fast does your method pick up on the collaborative aspect compared to a holistic version?
>
> The training dataset of our proposed method and M-Where2Act (the quadratic complexity version of Where2Act) are exactly the same, so the training data size is the same.
>
> To test the inference speed, for each method, we run 100 interaction trials and calculate the average time spent for proposing gripper actions. Our method spends **0.17 seconds** on average, while M-Where2Act spends **27.65 seconds** on average.
>
> The most difference in time consumption is in Affordance Network. Let $N$ denote the point number of the given point cloud observation ( $N=8192$ in our implementation). For our method, its two Affordance Networks sequentially propose the two affordance maps, so it should propose $N + N$ affordance scores in total, and the time complexity is $O(N)$. For M-Where2Act, its Affordance Network should propose the affordance score of all the $N * N$ point pairs, so the time complexity is $O(N^2)$ .
>
> > There are a few typos, e.g. in the Conclusion section ”[...] its superiority of the three baselines” should be ”over” etc.
>
> Thank you for pointing it out, and we have revised our paper.
>
> Thank you again for the valuable comments and questions, and we sincerely hope that our response above makes things clearer to you and addresses your concerns well.  Please feel free to let us know if you have further questions and we are happy to discuss further.
>
> * [Paper-1] Thomas Weng, Sujay Man Bajracharya, Yufei Wang, Khush Agrawal, and David Held. Fabricflownet: Bimanual cloth manipulation with a flow-based policy. In Conference on Robot Learning, pp. 192–202. PMLR, 2022.
>
> * [Paper-2] Samir Yitzhak Gadre, Kiana Ehsani, and Shuran Song. Act the part: Learning interaction strategies for articulated object part discovery. In Proceedings of the IEEE/CVF International Conference on Computer Vision (ICCV), pp. 15752–15761, October 2021.

---

> ### Author Response · Authors · 2022-11-16
> **REPLY: Thank you! We have addressed your concerns and revised our paper accordingly. Hope to hear back from you if you have further questions. [Part 1/2]**
>
> We sincerely thank you for your time and your positive feedback. Thank you for the valuable questions. We have addressed all the questions below and revised the paper accordingly to reflect the changes (we highlight all changes in red in the revised paper).
>
> > More discussion on Sec. 4.4 (Collaborative Adaption Procedure), ... It seems that the authors pair the learning networks with the robot to perform actions end-to-end in simulation, but how/why does that contribute more than training the components separately in an offline fashion?
>
> Thank you for this valuable question! It is true that "during Collaborative Adaptation (CA), we pair the learning networks with the robot to perform actions end-to-end in simulation" and "ground-truth offline data are generated with annotations for the desired collaboration".
>
> Even though ground-truth offline data are generated with annotations for the desired collaboration, the proposed affordance and actions may not be completely and perfectly consistent with the desired distributions of collected offline data, and will exist some errors. The errors of the two separately trained modules would accumulate and lead to even worse performance, and thus chances are that dual-gripper manipulation system will propose many actions (i.e., false positive actions) that cannot actually fulfill the tasks.
>
> Therefore, we propose the Collaborative Adaptation (CA) procedure, let two grippers perform actions end-to-end in simulation, and use the actual action results to adapt the dual-gripper manipulation system with the above explained cumulative errors. During this procedure, those false positive cases would be tuned to be negative, and thus the system will be adapted into a better collaborative system.
>
>
>
> > Comparison with other recent dual-gripper manipulation systems (e.g. Gadre et. al. 2021).
>
> Thanks for this valuable suggestion. We compare our method with **FabricFlowNet** (Weng et al., 2022;) [Paper-1] instead of **Act the Part** (Gadre et. al. 2021) [Paper-2]. In the following, we will first explain why we do not use Act the Part as a baseline, and then we will explain the experiment settings and results of FabricFlowNet in detail.
>
> _Why we do not compare with _**_Act the Part_**_._ The task of **Act the Part** is to segment parts of articulated objects (e.g. eyeglasses) from interactive perception. This method uses one gripper to hold one articulated part (e.g. one eyeglass leg), and uses the other gripper to move another part (e.g. another eyeglass leg). The observations before and after the interaction will be used to segment articulated parts. However, in our paper, we propose a generic framework that enables the two grippers to collaboratively accomplish different tasks for the target object, such as pushing or picking up a heavy box. Therefore, the tasks of the two works are totally different, our method is designed for collaboration, while Act the Part is designed for segmentation, so Act the Part is not suitable for our collaborative task. We are sorry that our Related Work Section does not clarify this difference clearly, and we have revised it.
>
> _Experiment details of _**_FabricFlowNet_**_ baseline_**_._ FabricFlowNet** uses two grippers to manipulate the cloth with a goal-conditioned flow-based policy. Given the initial and the target observations, FabricFlowNet can (1) use the FlowNet to predict the 2D optical flow, which represents the correspondence between two observations, (2) use the PickNet to take in the flow as input and predict the two pick points, (3) use the predicted flow at the pick points to find the two corresponding place points. Note that if the distance between pick points (or place points) is smaller than a threshold, they will use only a single gripper to manipulate to avoid the collision.
>
> In our implementation, (1) for optical flow, we calculate the ground-truth 3D flow between the initial observation and the target observation, and the ground-truth flow will be directly fed to the following PickNet. This means, we have the perfect flow. (2) For the PickNet, given the ground-truth flow, we adapt it from a 2D-version to a 3D-version by changing the backbone from FCN to PointNet++, and keeping other settings the same. (3) For the placing pose, since our task is in 3D space, apart from the place position, we also require the two grippers' movement direction. We design two methods to predict place pose: (i) using the flow direction at the pick point as the gripper's movement direction, and calculating the place position according to the flow; (ii) using our two Actor Networks to propose the two grippers' orientations, and calculating the place position according to the predicted orientations.

---

> ### Author Response · Authors · 2022-11-19
> **Looking Forward to Seeing Your Response!**
>
> Dear reviewer RF6q,
>
> Given the discussion phase is quickly passing, we want to know if our response resolves your concerns. If you have any further questions, we are more than happy to discuss them. Thanks again for your valuable suggestions!
>
> Best, All anonymous authors

---

> > ### Comment · Reviewer_RF6q · 2022-11-19
> > **feedback**
> >
> > Thank you for clarifying the concerns! It is an interesting read!

---

### Decision · Program_Chairs · 2023-01-20

**Decision:**

Accept: poster

**Justification For Why Not Higher Score:**

There are still concerns about the impact this work can have for the ICLR ML community.

**Justification For Why Not Lower Score:**

There are several novelties in this work that all the reviewers agree are valuable for the robot learning community.

**Metareview: Summary, Strengths And Weaknesses:**

This paper presents a new learning framework for dual gripper affordance for grasping-based manipulation tasks. We received four detailed  reviews from experts in robotics and machine learning. While the reviews were initially mixed, after discussion all reviewers agree that this work provides new insights into dual-arm manipulation from data collection to a new training strategy. The reviewers also brought up concerns about comparisons to non-affordance based baselines and the lack of discussion. However, we believe these will be addressed in the final version of this paper.

**Note From Pc:**

if the above contains the word "oral" or "spotlight" please see: "oral" presentation means -> notable-top-5% and "spotlight" means -> notable-top-25%. As stated in our emails, we are disassociating presentation type from AC recommendations